# LARGE LANGUAGE MODELS CAN BE STRONG DIFFERENTIALLY PRIVATE LEARNERS

**Xuechen Li**[1], **Florian Tramèr**[2], **Percy Liang**[1], **Tatsunori Hashimoto**[1]
[1]Stanford University [2]Google Research
{lxuechen,tramer,pliang}@cs.stanford.edu, thashim@stanford.edu

## ABSTRACT

Differentially Private (DP) learning has seen limited success for building large deep learning models of text, and straightforward attempts at applying Differentially Private Stochastic Gradient Descent (DP-SGD) to NLP tasks have resulted in large performance drops and high computational overhead. We show that this performance drop can be mitigated with (1) the use of large pretrained language models; (2) non-standard hyperparameters that suit DP optimization; and (3) fine-tuning objectives which are aligned with the pretraining procedure. With the above, we obtain NLP models that outperform state-of-the-art DP-trained models under the same privacy budget and strong non-private baselines—by directly fine-tuning pretrained models with DP optimization on moderately-sized corpora. To address the computational challenge of running DP-SGD with large Transformers, we propose a memory saving technique that allows clipping in DP-SGD to run without instantiating per-example gradients for any linear layer in the model. The technique enables privately training Transformers with almost the same memory cost as non-private training at a modest run-time overhead. Contrary to conventional wisdom that DP optimization fails at learning high-dimensional models (due to noise that scales with dimension) empirical results reveal that private learning with pretrained language models tends to not suffer from dimension-dependent performance degradation. Code to reproduce results can be found at https://github.com/lxuechen/private-transformers.

## 1 INTRODUCTION

Machine learning systems trained on sensitive user data can be vulnerable to privacy attacks (Shokri et al., 2017; Hayes et al., 2019). This issue is especially pressing for recent applications of large language models, as these models are capable of memorizing and reconstructing sensitive examples contained in the training data (Zhang et al., 2016; Carlini et al., 2020).

As a result of these concerns, there has been a large interest in developing methods that provide data privacy guarantees for large language models. The standard paradigm for providing such a guarantee in machine learning is *Differential Privacy* (DP) (Dwork et al., 2006; 2014). Unfortunately, DP learning has typically struggled to produce useful models when applied to large language models, resulting in models with either vacuous privacy guarantees (Dupuy et al., 2021) or performance far below non-private baselines. This is widely attributed to the fact that the core primitive of *Differentially Private Stochastic Gradient Descent* (DP-SGD) (Song et al., 2013; Bassily et al., 2014; Abadi et al., 2016) injects noise that must scale with the number of parameters, resulting in large noise levels for large language models (Yu et al., 2021b).

We tackle the problem of building performant DP language models for sentence classification and language generation tasks with merely tens to hundreds of thousands of examples. We pursue this goal by re-examining the performance of the baseline DP optimization algorithm for fine-tuning large language models, and study how choices of hyperparameters, training objective, and pretrained models affect the performance given fixed privacy budgets. In contrast to the mainstream perception, **our empirical results demonstrate that large pretrained models with hundreds of millions of parameters can be effectively and efficiently fine-tuned to yield models with high performance with modest privacy leakage.** For text generation, the performance of our models surpasses even strong non-private baselines. For sentence classification, the performance of our fine-tuned models

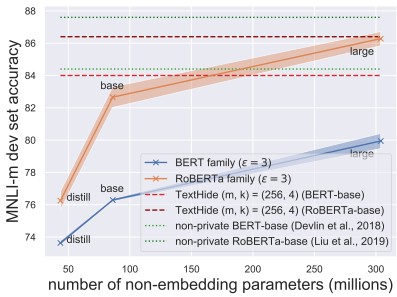

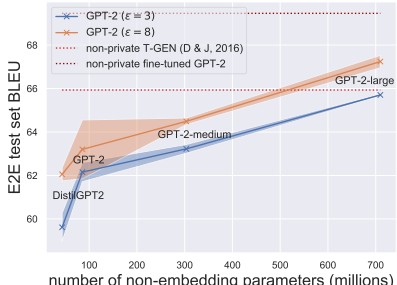

(a) Sentence classification
MNLI-matched (Williams et al., 2018)

(b) Natural language generation
E2E (Novikova et al., 2017)

Figure 1: A summary of a few of our findings: (1) Pretrained models fine-tuned with DP-Adam has strong performance. (2) Fine-tuning larger models produces better results. (3) Fine-tuned RoBERTa-large under DP at $\epsilon = 3$ outperforms TextHide (the extension of InstaHide (Huang et al., 2020b) for text classification) with BERT-base. Non-private generation baseline numbers are based on those reported by Wiseman et al. (2018).

surpasses those obtained under heuristic privacy notions (Huang et al., 2020a) which do not possess formal privacy guarantees. Figure 1 illustrates these results. We summarize our contributions below.

(1) We show that with appropriate hyperparameters and downstream task objectives, fine-tuning pretrained language models with DP-SGD/DP-Adam yields strong performance for a suite of NLP tasks at privacy levels $\epsilon \in \{3, 8\}$. Some of our fine-tuned models outperform strong non-private learning baselines and models obtained under heuristic privacy notions.

(2) Running DP-SGD can be memory-intensive due to clipping per-example gradients. We present *ghost clipping*, a memory saving technique that makes fine-tuning large Transformers under DP memory efficient. Our technique generalizes the Goodfellow (2015) trick to handle sequential inputs, and can be combined with a layer-by-layer clipping procedure (Lee & Kifer, 2020) to enable privately fitting large Transformers with almost the same memory cost as non-private training—at the cost of one additional backward pass per processed batch.

(3) We show that the dimensionality of gradient updates does not explain private fine-tuning performance. While there exist dimension-dependent lower bounds for private (convex) optimization (Bassily et al., 2014), we find that larger pretrained models lead to better private fine-tuning results. Moreover, parameter-efficient adaptation methods that reduce the dimensionality of updates do not necessarily outperform a baseline method that fine-tunes all model parameters.

Our empirical studies indicate that directly fine-tuning pretrained models with DP optimization results in performant DP language models under modest privacy budgets. This enables building practical private NLP models for a range of common tasks where privacy could be at stake.

## 2    PROBLEM STATEMENT

We build models for sentence classification and language generation tasks with datasets of modest sizes under (central/global) approximate-DP (also known as $(\epsilon, \delta)$-DP) (Dwork et al., 2014).

**Definition 1** ($(\epsilon, \delta)$-DP). *A randomized algorithm $\mathcal{M} : \mathcal{X} \to \mathcal{Y}$ is $(\epsilon, \delta)$-differentially private if for all adjacent datasets $X, X' \in \mathcal{X}$ and all $Y \subset \mathcal{Y}$, $\mathbb{P}(\mathcal{M}(X) \in Y) \leq e^{\epsilon}\mathbb{P}(\mathcal{M}(X') \in Y) + \delta$.*

Two datasets are adjacent if and only if one can be obtained from the other by including an extra record (Mironov et al., 2019).[1] How a record is defined is task dependent and will be made clear below. Intuitively, DP algorithms ensure that random outputs obtained from similar inputs are difficult to distinguish. $\epsilon$ and $\delta$ are *privacy leakage* parameters that measure the loss of privacy and small values imply more privacy. Unlike heuristic privacy notions (Huang et al., 2020b), DP allows for the tracking of privacy loss through the calculation of leakage parameters, and ensures privacy under composition (Dwork et al., 2014), meaning that the overall privacy loss of multiple DP algorithms releasing multiple statistics can be reasoned in a principled manner.

---

[1]An alternative definition of adjacency assumes all datasets are of equal size and relies on the replacement of records. This is not the definition we adopt.

DP learning typically relies on DP optimizers which privatize gradients before performing updates. The privatization step ensures that the parameter updates leak limited information about the training examples through their gradients. Specifically, this step clips per-example gradients with a norm constraint $C$, and adds Gaussian noise $z \sim \mathcal{N}(0, C^2\sigma^2 I_p)$ to the sum of clipped gradients. Here, $\sigma$ is the *noise multiplier* determined from the privacy budget $(\epsilon, \delta)$, number of gradient updates $S$, and sampling rate $q = \frac{B}{N}$ for a batch size of $B$ and a dataset with $N$ examples. Intuitively, clipping individual gradients ensures that each example has bounded influence on the parameter update, whereas noising the gradient prevents exact tracing of particular examples. The noise being isotropic implies that larger models would experience heavier noise per update, as the norm of the $p$-dimensional Gaussian $\|z\|_2$ scales as $C\sigma\sqrt{p}$. This is widely believed to be the cause for DP optimization performing poorly at training high-dimensional deep learning models (Kamath, 2020).

Our starting point for building DP language models is (public) pretrained models. Pretrained language models tend to contain general knowledge of language (Manning et al., 2020) and thus should make the downstream private learning problem easier. We fine-tune these models with DP-Adam (Abadi et al., 2016; Kingma & Ba, 2014) (see Appendix A for details) and track privacy loss through Rényi DP (Mironov, 2017), but also report the converted $\epsilon$ from a Gaussian DP CLT (Dong et al., 2019) and from accurately composing tradeoff functions via fast Fourier transform (Gopi et al., 2021). We consider privacy levels $\epsilon \in \{3, 8\}$ and $\delta = 1/2|\mathcal{D}_{\text{train}}|$ throughout for a training set of size $|\mathcal{D}_{\text{train}}|$ (see Appendix B for further details). We tune hyperparameters on a text generation task (E2E; introduced below) and transfer these to remaining tasks. We outline two broad classes of NLP problems considered in this paper and define what constitutes a record below.

**Sentence Classification.** The goal is to learn a model that classifies sentences into one of a few categories. For these tasks, each example/record consists of input sentences and a label to be predicted. We fine-tune models of various sizes in the BERT (Devlin et al., 2018) and RoBERTa (Liu et al., 2019) families, as these models are known to work well for sentence classification.

**Language Generation.** The goal is to learn a model that generates natural language sentences given some context. For table-to-text generation tasks such as E2E (Novikova et al., 2017) and DART (Nan et al., 2020), each example/record in the training data consists of a pair of table entry and corresponding text description to be predicted. For a dialogue generation task such as Persona-Chat (Zhang et al., 2018), each example/record consists of metadata, a dialogue history, and a response to be predicted. We fine-tune GPT-2 (Radford et al., 2019) of various sizes for these problems, as this model family is known to work well for text generation.

## 3 EFFECTIVE DIFFERENTIALLY PRIVATE FINE-TUNING

By studying the impact of hyperparameters and choice of fine-tuning objective, we demonstrate that the performance of the DP-Adam baseline can be substantially improved, even matching some strong non-private learning results. Our analyses reveal common failure modes when straightforwardly applying DP optimization and explain poor results reported in past works that consider these baselines.

### 3.1 HYPERPARAMETER TUNING

DP optimization is sensitive to the choice of hyperparameters (Papernot et al., 2019). Our experiments suggest that performance can vary from being close to trivial with ill-chosen hyperparameters to near previous state-of-the-art with appropriately chosen ones. As a consequence, we present simple but effective guidelines on setting the most important hyperparameters. Unless otherwise stated, the unmentioned hyperparameters are set to defaults documented in Appendix H.

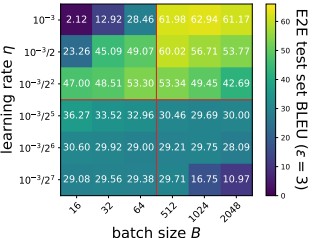

### 3.1.1 BATCH SIZE, LEARNING RATE & TRAINING EPOCHS

Our experiments suggest that batch size is one of the most important hyperparameters to set correctly, and the dependence of the optimal batch size on learning rate and training epochs makes its selection complex. We first describe batch size selection in realistic, compute-bound settings and then describe how the complexity of identifying the optimal batch size in these situations arise due to constraints on the number of training epochs.

Figure 2: Large batch sizes and learning rates lead to the best performance when $E$ is fixed. Red lines divide heat map into four panels. Top and bottom correspond to low and high learning rate regimes; left and right correspond to small and large batch regimes.

**Fixed Training Epochs** $E$. We first describe a realistic situation in which there is a constraint on the total amount of compute budget available. For the case of DP-SGD, this compute budget constraint often corresponds to a constraint on the number of examples that are processed by SGD.[2] In this fixed training epoch setting, the learning rate and batch size jointly affect performance, since using larger batches implies performing fewer gradient updates. To study this joint influence empirically, we fine-tune GPT-2 on the E2E dataset for table-to-text generation with DP-Adam at $\epsilon = 3$ with various batch sizes and learning rates. Figure 2 shows that the best performing models are obtained with both a large batch size and large learning rate. Using a small learning rate together with a small batch size yields considerably worse results. Note a seq2seq baseline achieves a test BLEU of ~65 without privacy here (Wiseman et al., 2018).

Recall that in the non-private world, pretrained language models are typically fine-tuned with small batch sizes and small learning rates with Adam (bottom left panel in Figure 2). This implies that naïely fine-tuning pretrained language models privately using the non-private setup would result in more performance degradation than necessary.

Recently, Tramèr & Boneh (2020) studied how the batch size and learning rate jointly affect learning private image classifiers while holding other hyperparameters fixed. They heuristically suggested a *linear scaling rule*: Scaling the learning rate together with the batch size by the same constant should yield models with almost the same performance. However, Figure 2 indicates that this fails to hold consistently as it falsely predicts that large batch and high learning rate (top right most entry) would have equal performance to small batch and low learning rate (bottom left entry). We further explain why linear scaling fails for small batches in Appendix D.

**Fixed Update Steps** $S$. In the fixed epoch setting, we saw that the optimal batch size setting was complex due to the trade-off between batch size and number of gradient updates. We now show that the complexity of setting batch sizes arises almost entirely from this tradeoff by considering a different setting, where the total number of gradient updates (rather than epochs) is fixed. In this case, using larger batches implies training for more epochs, and we find that using larger batch sizes almost always results in better performance at a given privacy budget (at the cost of processing more examples and using more compute), once other hyperparameters are fixed. We provide an explanation of this by introducing the idea of an *effective noise multiplier* $\sigma_{\text{eff}} = \frac{\sigma}{q} = \frac{\sigma N}{B}$. Recall the noise multiplier $\sigma$ is determined from the privacy budget $(\epsilon, \delta)$, update steps $S$, and sampling rate $q$. In addition, recall the form of the privatized gradient $\bar{g}$ in DP-SGD/DP-Adam:

$$\bar{g} = \widetilde{g} + \bar{z}, \quad \widetilde{g} = \frac{1}{B}\sum_{i=1}^{B}\text{Clip}(\nabla\mathcal{L}_i, C), \quad \bar{z} \sim \mathcal{N}\left(0, C^2\frac{\sigma^2}{B^2}I_p\right) = \mathcal{N}\left(0, C^2\frac{\sigma_{\text{eff}}^2}{N^2}I_p\right),$$

where $\nabla\mathcal{L}_i$ is the gradient of the $i$th example in a batch of $B$ examples and $\text{Clip}(v, K)$ clips the vector $v$ by the norm constraint $K$. We observe that for moderately large batches, the signal-to-noise ratio $r = \|\widetilde{g}\|_2/\|\bar{z}\|_2$ is mainly controlled by the batch size through the effective noise multiplier: The signal term $\widetilde{g}$ tends to concentrate quickly due to being an average of bounded vectors, whereas the effective noise multiplier $\sigma_{\text{eff}}$ could vary significantly as the batch size $B$ changes. Figure 3 (a) shows that the effective noise multiplier decreases as batch size increases. In addition, Figure 3 (b) plots the average signal-to-noise ratio $\bar{r}$ over the first 30 gradient updates against the final model's performance on E2E and demonstrates that large batches (up to a threshold) lead to both increased signal-to-noise ratio at the beginning of training and better performance at the end of training. These findings additionally resonate with and explains recent empirical successes of private pretraining (Anil et al., 2021).

### 3.1.2 CLIPPING NORM

DP optimization is known to be sensitive to the choice of clipping norm. Since the scale of noise depends on this clipping norm (recall its standard deviation is $C\sigma$), picking the threshold $C$ much larger than the actual gradient norm implies more noise is being applied than necessary. In practice, we have found that a small clipping norm which enforces almost all gradients to be clipped throughout training leads to the best performing models (see Figure 8 in Appendix H).

### 3.2 IMPROVING THE TASK ALIGNMENT HELPS PRIVATE LEARNING

Our fine-tuned models on language generation tasks worked well since the pretraining objective and downstream task are *aligned*: Both involve predicting sequences of tokens. This alignment simplified

---

[2]This is because DP-SGD often necessitates microbatching, in which case the number of backward passes is independent of the actual batch size for gradient updates but dependent on numbers of passes through the data.

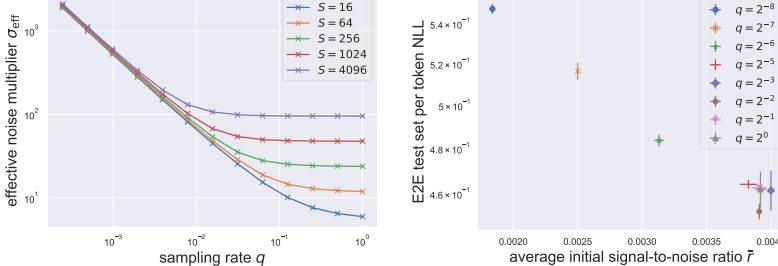

Figure 3: **Left:** Effective noise multiplier decreases with increasing sampling rate for various fixed $S$. **Right:** Large batch sizes (corresponding to large $q$ in the figure) have higher signal-to-noise ratio, which (log-)linearly correlates with final model performance.

the task and benefitted private learning. While pretrained models are naturally aligned for language generation, it is much less so for classification tasks. The standard approach for adapting language models for classification involves stacking a freshly initialized network on top of the encoding of the special `[CLS]` token and jointly optimizing all parameters (Devlin et al., 2018). This workflow introduces a discrepancy between pretraining and fine-tuning: Pretraining predicts masked out words from a large vocabulary whereas fine-tuning predicts integer labels.

To eliminate the discrepancy, we instead consider learning to predict the missing word during fine-tuning for classification. For example, for sentiment classification, we reframe the problem as filling in the `[MASK]` token in the sequence "<INPUT>. It is `[MASK]`." and compare the probabilities of words "awesome" and "terrible". This text infilling task is almost exactly the procedure used for pretraining masked language models, and recent works have demonstrated its effectiveness for knowledge probing (Petroni et al., 2019), few-shot learning (Gao et al., 2020) and multi-task fine-tuning (Wei et al., 2021). On SST-2, we found that using the generic template as described above already improved private fine-tuning performance by $3 \sim 5\%$ across different settings. Table 1 (to be presented) contains additional results and Appendix E includes analyses on choices of label words.

## 4 GHOST CLIPPING: CLIPPING WITHOUT PER-EXAMPLE GRADIENTS

DP-SGD has high memory overhead due to clipping *per-example* gradients. Naïvely implemented, this step instantiates a giant gradient vector for each example during optimization and can be prohibitively expensive. For example, Hoory et al. (2021) pretrained BERT with DP optimization and reported memory issues when using the large batches necessary to achieve high performance. A time-costly solution to the memory problem is micro-batching: Split large batches into multiple smaller ones and aggregate the results after processing each small batch individually (Tramèr & Boneh, 2020). This solution, however, is unlikely to be sufficient as neural language models become larger and fitting a few copies of the gradient in memory can be difficult. Lee & Kifer (2020) observed that per-example gradients need not be instantiated at all, if the goal is to sum the clipped gradients. They presented a clipping procedure that only instantiates the per-example gradient for parameters of a *single* layer in the model one at a time, as opposed to the entire model at once, at the cost of an extra backpropagation pass per processed batch.

Unfortunately, we find this trick to be still insufficient for sequence models such as Transformers (Vaswani et al., 2017), as the memory requirement for per-example gradients of embedding layers and language modeling heads can be costly. We extend the Lee & Kifer (2020) approach such that training Transformers with DP optimization can have almost the same memory consumption as non-private training. Unlike their approach, our extension avoids instantiating the per-example gradient even for individual linear layers. We call this approach *ghost clipping*, as the per-example gradient is the ghost that never explicitly appears. We anticipate this extension to be useful for both privately fine-tuning and pretraining large Transformers.

### 4.1 THE MEMORY TRICK BY LEE & KIFER (2020)

Per-example gradient clipping is easy if we know per-example gradient norms. In this case, we first compute the scaling factor $c_i = \min(1, C/\|\nabla\mathcal{L}_i\|_2)$, where $C$ is the clipping threshold and $\mathcal{L}_i$ is the loss associated with the $i$th example. Then, we perform the usual backward pass with the reweighted

scalar loss $\sum_i c_i \mathcal{L}_i$. This procedure gives us the sum of clipped gradients. Under this setup, the difficulty is computing the per-example gradient norm $\|\nabla \mathcal{L}_i\|_2$. We emphasize two technicalities that enable computing this quantity without instantiating the full per-example gradient $\nabla \mathcal{L}_i$.

First, for a typical neural net layer $l$ with parameters $W^{(l)}$ (without parameter sharing), the per-example gradient w.r.t. parameters can be easily computed using the input to the layer $a^{(l)}$ and the gradient of the loss w.r.t. the output $g^{(l)}$, both of which are available during backpropagation. Second, for a large vector formed by concatenating several small vectors $u = [u_1, \ldots, u_k]$, its Euclidean norm is simply the norm of the vector of norms, i.e. $\|u\|_2 = \|(\|u_1\|_2, \ldots, \|u_k\|_2)\|_2$. The second observation means that computing the per-example gradient norm $\|\nabla \mathcal{L}_i\|_2$ can be done by computing the per-example gradient norms for individual layers of the neural net $\|\nabla_{W^{(1)}} \mathcal{L}_i\|_2, \ldots, \|\nabla_{W^{(L)}} \mathcal{L}_i\|_2$ one at a time ($L$ is layer count). Moreover, the first observation implies that the norms for each layer can be computed using quantities freely available to a typical backward pass. Overall, the per-example gradient norm of any network without parameter sharing can be computed in a layer-by-layer fashion with only one per-example gradient tensor for a single layer being instantiated at any time.

## 4.2 GHOST CLIPPING FOR TRANSFORMERS WITH SEQUENTIAL DATA

The trick by Lee & Kifer (2020) still requires instantiating the per-example gradient of individual layers (although not simultaneously). This can be problematic in terms of memory for Transformers with large embedding layers.[3] Here, we present a specialized procedure for computing the per-example gradient norm for linear and embedding layers when they are applied to sequential data.[4] This procedure reduces memory footprint and can be viewed as a generalization of the Goodfellow (2015) trick that additionally handles sequential inputs.

Let $a \in \mathbb{R}^{B \times T \times d}$ be the input to a linear layer with weight matrix $W \in \mathbb{R}^{p \times d}$, and $s \in \mathbb{R}^{B \times T \times p}$ be the output with $s_{i,j} = W a_{i,j}$. Let $g \in \mathbb{R}^{B \times T \times p}$ be the gradient of the loss w.r.t. the output $s$. Here, $T$ is the number of time steps in the input, and we omitted biases for simplicity. Simple calculation shows that the per-example gradient is the product of two matrices:

$$\nabla_W \mathcal{L}_i = g_i^\top a_i \in \mathbb{R}^{p \times d}. \tag{1}$$

Since the per-example gradient norms are the end goal, the per-example gradients $\{\nabla_W \mathcal{L}_i\}_{i=1}^B$ themselves need not be instantiated explicitly. More precisely, we observe that the squared per-example gradient norm for this layer $\|\nabla_W \mathcal{L}_i\|_F^2$ obeys the following identity:

$$\|\nabla_W \mathcal{L}_i\|_F^2 = \text{vec}(a_i a_i^\top)^\top \text{vec}(g_i g_i^\top). \tag{2}$$

See Appendix F for a derivation. Implemented with common primitives in machine learning libraries, (2) has a memory complexity of order $\mathcal{O}(BT^2)$ when $a_i a_i^\top, g_i g_i^\top \in \mathbb{R}^{T \times T}$ are instantiated,[5][6] as opposed to $\mathcal{O}(Bpd)$ in the naïve approach which goes through instantiating (1).[7]

The memory efficiency of this procedure is exemplified with off the shelf pretrained language models, most of which have large embeddings. For instance, for GPT-2, $d \approx 50,000$ and $p = 768$ for the embedding layer, and the context window $T \leq 1024$.[8] Our method in theory reduces the memory cost of this large embedding layer by at least a factor of 22. In practice, we also observe significant savings, since embedding layers can be a major source of memory spending for training large language models.[9] To stress-test ghost clipping, we compare it with 4 baselines: The `PyTorch` package `Opacus` that implements DP optimization by instantiating per-example gradients, the approach by Lee & Kifer (2020), non-private training in `PyTorch`, and naïve DP optimization implemented in

---

[3]For GPT-2, per-example gradients w.r.t. the embedding for ten examples alone occupy ~1.5GB of memory.

[4]An embedding layer is essentially a linear layer: The embedding lookup operation applied to indices is equivalent to a matrix multiplication of the embedding matrix with one-hot encoded indices.

[5]This is assuming the space complexity for multiplying two matrices $A \in \mathbb{R}^{m \times n}$ and $B \in \mathbb{R}^{n \times p}$ is roughly $\mathcal{O}(mp)$, which is the case for most workloads running on a framework like PyTorch.

[6]More sophisticated solutions may even avoid instantiating $a_i a_i^\top$ and $g_i g_i^\top$ entirely by trading in more run-time. Custom CUDA kernels are likely needed to make these solutions fast in practice.

[7]We omitted the cost of storing $a_i$ and $g_i$, since our goal is to compare the additional cost induced by computing gradient norms.

[8]In practice, for fine-tuning tasks, the maximum sequence length is usually a few hundred.

[9]While there are alternative approaches for reducing the memory footprint of embedding layers during training, these methods tend to introduce extra hyperparameters that require tuning and privacy spending.

JAX with `jit` and `vmap` enabled. We include the `JAX` baseline since recent studies show that DP optimization can be made cheap through compiler optimization (Subramani et al., 2020). Figure 4 (a) shows that for typical inputs, our technique is the most memory friendly and allows fitting batches almost as large as those in non-private training. Since ghost clipping allows us to fit larger batches but with a run-time penalty, a natural question is whether it improves throughput with the use of larger batches. Figure 4 (b) shows that while ghost clipping only provides minor gains compared to `Opacus` for smaller models, it allows processing ~10% more examples compared to the approach by Lee & Kifer (2020) for fitting GPT-2-large, a model that neither `Opacus` or `JAX` could handle. See Appendix G for the setup of these experiments.

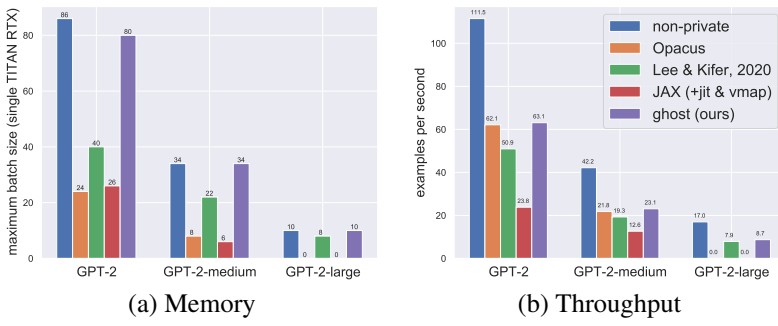

(a) Memory  (b) Throughput

Figure 4: **Left:** Ghost clipping is 3 times more memory efficient than `Opacus` and is almost as efficient as non-private training for typical sequences across model sizes. For GPT-2-large, we were unable to fit single-example micro batches together with gradient accumulation with `Opacus` or `JAX` on a TITAN RTX GPU (24 GBs of VRAM). **Right:** DP optimization with ghost clipping processes ~10% more examples than the approach by Lee & Kifer (2020) under unit time for GPT-2-large.

## 5  LOW DIMENSIONAL UPDATES ARE NOT NECESSARILY BETTER

Since the norm of the noise added to gradients in DP scales with dimensionality, it is natural to ask whether privatizing and updating fewer parameters would result in improved performance. We decompose this question into two aspects: (1) Do smaller pretrained models lead to better private fine-tuned performance, and (2) do parameter-efficient adaptation methods designed with a reduced dimensionality of updates outperform full fine-tuning? Our experiments below show that the answer to both questions is negative. Reported numbers in this section are averaged over three seeds.

### 5.1  LARGER PRETRAINED MODELS RESULT IN BETTER PERFORMANCE

We observe that larger pretrained models lead to better private fine-tuned performance. Specifically, we fully fine-tune four sizes of GPT-2 models (for language generation) and three sizes of BERT/RoBERTa models (for sentence classification) at the same privacy budget with DP-Adam and compare their performances. Since the performance of DP optimization heavily depends on hyperparameter choices, we need to ensure that our hyperparameters are not particularly favoring larger models. We thus tune hyperparameters on the smallest model and then reuse the same hyperparameters for all fine-tuning workloads. Figure 1 from earlier demonstrates gains on E2E and MNLI from model scaling, and we find similar improvements on 5 additional tasks (deferred to Figure 5 in Appendix C).

### 5.2  FULL FINE-TUNING WITH DP-ADAM MATCHES STATE-OF-THE-ART

There is a range of lightweight fine-tuning methods that reduce the dimensionality of updates, including some that are designed for DP (Yu et al., 2021c). Do methods that optimize fewer parameters lead to better results under DP even if they perform similarly non-privately? Empirical results suggest otherwise and that full fine-tuning is a strong baseline that even matches specialized low-dimensional DP learning methods for both classification and generation. Below, we study the two sets of tasks separately. For completeness, all experimental details are in Appendix K.

**Sentence Classification.**  We study DP fine-tuning on tasks from the GLUE benchmark that have more than 10k training examples (MNLI, QQP, QNLI, and SST-2), following the experimental setup of Yu et al. (2021c). The associated datasets have modest sizes: SST-2 and QNLI have 60k+ and 100k+ training examples, respectively. MNLI and QQP each contains less than 400k examples.

Table 1 shows that using larger pretrained models and the text-infilling objective generally improve classification accuracy. We compare full fine-tuning with *reparameterized gradient perturbation* (RGP) (Yu et al., 2021c), as it is the state-of-the-art for DP fine-tuning on sentence classification at the time of writing. The method is designed to privatize gradients projected onto low dimensional subspaces and was motivated to reduce DP noise in high-dimensional models. We note that full fine-tuning with the text infilling objective outperforms well-tuned RGP on all tasks despite being the simplest baseline. Computationally, while RGP is faster per-update, it requires more than 3 times as many epochs as full fine-tuning – overall, both methods comparable in terms of wall time.

Table 1: Full fine-tuning larger pretrained models with text infilling has best performance. Results are dev set accuracies. Best numbers based on two-sample test for each privacy level are in bold.

| Method | $\epsilon = 3$ | | | | $\epsilon = 8$ | | | |
| | MNLI-(m/mm) | QQP | QNLI | SST-2 | MNLI-(m/mm) | QQP | QNLI | SST-2 |
|---|---|---|---|---|---|---|---|---|
| RGP (RoBERTa-base) | - | - | - | - | 80.5/79.6 | 85.5 | 87.2 | 91.6 |
| RGP (RoBERTa-large) | - | - | - | - | 86.1/86.0 | 86.7 | 90.0 | 93.0 |
| full (RoBERTa-base) | 82.47/82.10 | 85.41 | 84.62 | 86.12 | 83.30/83.13 | 86.15 | 84.81 | 85.89 |
| full (RoBERTa-large) | 85.53/85.81 | **86.65** | 88.94 | 90.71 | 86.28/86.54 | **87.49** | 89.42 | 90.94 |
| full + infilling (RoBERTa-base) | 82.45/82.99 | 85.56 | 87.42 | 91.86 | 83.20/83.46 | 86.08 | 87.94 | 92.09 |
| full + infilling (RoBERTa-large) | **86.43/86.46** | 86.43 | **90.76** | **93.04** | **87.02/87.26** | 87.47 | **91.10** | **93.81** |
| $\epsilon \approx$ (Gaussian DP + CLT) | 2.52 | 2.52 | 2.00 | 1.73 | 5.83 | 5.85 | 4.75 | 4.33 |
| $\epsilon \approx$ (Compose tradeoff func.) | 2.75 | 2.75 | 2.57 | 2.41 | 7.15 | 7.16 | 6.87 | 6.69 |

**Table-To-Text Generation.** We study different fine-tuning methods under DP for table-to-text generation where the goal is to generate natural language descriptions of table entries. We consider the datasets E2E (Novikova et al., 2017) and DART (Nan et al., 2020). E2E consists of simple restaurant reviews, whereas DART consists of open-domain table entries from Wikipedia and is more complex. Both datasets are small: E2E has more than 40k training examples, whereas DART has more than 60k. Since we are the first to experiment with this task under DP, we compare full fine-tuning (full) against a suite of parameter-efficient approaches which includes LoRA (Hu et al., 2021), prefix-tuning (Li & Liang, 2021) (prefix), RGP, and fine-tuning the top 2 Transformer blocks (top2), all of which optimize few parameters. On GPT-2 (125 million parameters), prefix-tuning with default hyperparameters optimizes ~10 million parameters; LoRA with rank 4 optimizes ~0.15 million parameters. We also report results for training from randomly initialized weights (retrain). Hyperparameters of each method were tuned only the E2E dataset; the complete search ranges are in Appendix I. Table 2 shows that LoRA and full fine-tuning are generally the most performant on E2E. Tables 7 and 8 in Appendix J contain the full result on E2E and DART and confirm the trend.

Table 2: Full fine-tuning performs on par with or outperforms others methods that execute gradient update in low dimensional spaces. Results are on E2E from fine-tuning GPT-2.

| Metric | DP Guarantee | Gaussian DP + CLT | Compose tradeoff func. | Method | | | | | |
| | | | | full | LoRA | prefix | RGP | top2 | retrain |
|---|---|---|---|---|---|---|---|---|---|
| BLEU | $\epsilon = 3$ | $\epsilon \approx 2.68$ | $\epsilon \approx 2.75$ | **61.519** | 58.153 | 47.772 | 58.482 | 25.920 | 15.457 |
| | $\epsilon = 8$ | $\epsilon \approx 6.77$ | $\epsilon \approx 7.27$ | **63.189** | 63.389 | 49.263 | 58.455 | 26.885 | 24.247 |
| | non-private | - | - | 69.463 | 69.682 | 68.845 | 68.328 | 65.752 | 65.731 |
| ROUGE-L | $\epsilon = 3$ | $\epsilon \approx 2.68$ | $\epsilon \approx 2.75$ | 65.670 | 65.773 | 58.964 | 65.560 | 44.536 | 35.240 |
| | $\epsilon = 8$ | $\epsilon \approx 6.77$ | $\epsilon \approx 7.27$ | 66.429 | 67.525 | 60.730 | 65.030 | 46.421 | 39.951 |
| | non-private | - | - | 71.359 | 71.709 | 70.805 | 68.844 | 68.704 | 68.751 |

**Chit-Chat Dialog Generation.** We stress-test full fine-tuning under DP with the task of chit-chat dialog generation. This task has the distinct challenge that the response space is intrinsically diverse (Li et al., 2015; Gao et al., 2018) since human conversations can be informal and noisy (Zhang et al., 2019). Moreover, dialog datasets are usually formed with user data which may contain sensitive information. We use the Persona-Chat dataset (Zhang et al., 2018) as a testbed and build off a processed version that has ~130k training entries. Each entry contains a dialog history, persona

descriptions of the respondent, and the response. We fine-tune GPT-2, GPT2-medium, and DialoGPT-medium on this dataset both privately and non-privately by training to predict the response with the dialog history and persona description. We report the F1 score and perplexity on the validation split, and human evaluated quality scores of generations. Table 3 shows that private models have strong performance. In particular, fine-tuned DialoGPT-medium at $\epsilon = 8$ beats the (non-private) winning entry of the ConvAI2 challenge (Dinan et al., 2019) on perplexity and has a human evaluation rating that is close to non-private models. Samples from our private models can be found in Appendix O.

Table 3: Fine-tuning with DP-Adam yields high quality chit-chat dialog generation models.

| Model | DP Guarantee | Gaussian DP +CLT | Compose tradeoff func. | Metrics | | |
|---|---|---|---|---|---|---|
| | | | | F1 ↑ | Perplexity ↓ | Quality (human) ↑ |
| GPT-2 | $\epsilon = 3$ | $\epsilon \approx 2.54$ | $\epsilon \approx 2.73$ | 15.90 | 24.59 | - |
| | $\epsilon = 8$ | $\epsilon \approx 6.00$ | $\epsilon \approx 7.13$ | 16.08 | 23.57 | - |
| | non-private | - | - | 17.96 | 18.52 | - |
| GPT-2-medium | $\epsilon = 3$ | $\epsilon \approx 2.54$ | $\epsilon \approx 2.73$ | 15.99 | 20.68 | - |
| | $\epsilon = 8$ | $\epsilon \approx 6.00$ | $\epsilon \approx 7.13$ | 16.53 | 19.25 | - |
| | non-private | - | - | 18.64 | 15.40 | - |
| DialoGPT-medium | $\epsilon = 3$ | $\epsilon \approx 2.54$ | $\epsilon \approx 2.73$ | **17.37** | **17.64** | 2.82 (2.56, 3.09) |
| | $\epsilon = 8$ | $\epsilon \approx 6.00$ | $\epsilon \approx 7.13$ | **17.56** | **16.79** | 3.09 (2.83, 3.35) |
| | non-private | - | - | 19.28 | 14.28 | 3.26 (3.00, 3.51) |
| HuggingFace (ConvAI2 winner) | non-private | - | - | 19.09 | 17.51 | - |
| HuggingFace (our implementation) | non-private | - | - | 16.36 | 20.55 | 3.23 (2.98, 3.49) |
| Reference | - | - | - | - | - | 3.74 (3.49, 4.00) |

## 6 RELATED WORK

**Private NLP.** The privacy-preserving NLP space is largely divided by whether or not a formal notion of privacy is considered. McMahan et al. (2017) successfully trained small word-level RNNs with 1.35 million parameters in a federated setting with more than 700k users under a global DP guarantee of $(\epsilon, \delta) = (4.6, 10^{-9})$. Ramaswamy et al. (2020) train production grade next-word prediction models using DP-FedAvg with millions of users. Qu et al. (2021) studied fine-tuning BERT for language understanding tasks under local DP. Kerrigan et al. (2020) presented initial results that public pretraining is helpful for downstream DP fine-tuning. However, they did not attempt fine-tuning large pretrained models with DP-SGD. Bommasani et al. (2021) briefly commented on the possibility of achieving cheaper private learning by fine-tuning large pretrained language models. Anil et al. (2021) pretrained BERT under global DP on datasets with hundreds of millions of examples. Dupuy et al. (2021) studied private BERT fine-tuning on datasets of utterances, but reported results with $\epsilon$ on the order of at least 100. Orthogonally, many works considered training language models that satisfy empirical notions of privacy (Xu et al., 2021; Coavoux et al., 2018; Mireshghallah et al., 2021; Melamud & Shivade, 2019). Our work is distinct from all works mentioned above in that we study fine-tuning large language models (with hundreds of millions of parameters) under global DP with stringent guarantees ($\epsilon \in \{3, 8\}$) on smaller datasets (much less than a million examples).

## 7 SCOPE AND LIMITATIONS

We presented strategies for fine-tuning large pretrained language models under DP for a wide range of NLP tasks. For researchers and practitioners working on private NLP, our empirical results suggest that DP fine-tuning with a proper setup is a competitive baseline that is worth trying before prematurely shifting to less formal notions of privacy which have not stood against the test of time. Below we list some limitations and future directions.

**Pretraining vs Private Learning.** Our model scaling results suggest that using larger pretrained models improves performance. This argument, however, is dependent on the particular choice of pretrained models. How pretraining helps private learning and whether better pretrained models for private learning could be built are interesting future avenues.

**Scaling Laws for Private Learning.** While scaling laws (Kaplan et al., 2020) for non-private deep learning have become prevalent, we are unaware of a case study in the private realm. Studies on how the dimensionality of models (and pretraining) generally affect private deep learning in precise quantitative terms will likely be a useful tool in trading off compute budget and model quality.

ETHICS STATEMENT

While the present paper studies private NLP, all experiments performed herein are based on publicly available datasets for reproducibility purposes. The paper empirically studies various behaviors of (mostly existing) algorithms, thus likely does not introduce any new ethical or societal concerns. The present paper does not introduce any new dataset.

REPRODUCIBILITY

All experiments in the paper are based on publicly available datasets. Links to these datasets are included in the main text and appendices. Hyperparameters necessary for reproducing our experiments are documented in Appendix I and H. Code for the ghost clipping technique can be found at https://github.com/lxuechen/private-transformers.

ACKNOWLEDGMENTS

We thank Xiang Lisa Li for help on reproducing non-private prefix-tuning results. We thank Da Yu for discussions and help on reproducing results for the RGP method. We thank Abhradeep Guha Thakurta and Rishi Bommasani for helpful discussions. We thank members of the Stanford statistical machine learning group for comments on early versions of the abstract. We thank Guodong Zhang and Mayee Chen for comments on an early draft. We thank Stanford HAI for a Google Cloud Credits Grant. XL is supported by a Stanford Graduate Fellowship.

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

## A  DP-ADAM

We use DP-Adam throughout. DP-Adam works just like regular Adam (Kingma & Ba, 2014) but performs updates and moment accumulation with privatized gradients. The gradient privatization part is the same as that performed in DP-SGD (Song et al., 2013; Abadi et al., 2016). Seemingly uncommon, DP-Adam is used in many private ML libraries.[10] To determine the noise multiplier, we account privacy through Rényi differential privacy (RDP) (Mironov, 2017; Mironov et al., 2019). For completeness, we include the pseudocode for DP-Adam below.

---

**Algorithm 1** DP-Adam

---

1: **Input:** Data $\mathcal{D} = \{x_i\}_{i=1}^N$, learning rate $\eta$, noise multiplier $\sigma$, batch size $B$, Euclidean norm threshold for gradients $C$, epochs $E$, initial parameter vector $\theta_0 \in \mathbb{R}^p$, initial moment estimates $m_0, v_0 \in \mathbb{R}^p$, exponential decay rates $\beta_1, \beta_2 \in \mathbb{R}$, avoid division-by-zero constant $\gamma \in \mathbb{R}$.
2: **for** $t \in [E \cdot N/B]$ **do**
3:     Draw a batch $B_t$ via Poisson sampling; each element has probability $B/N$ of being selected
4:     **for** $x_i \in B_t$ **do**
5:         $g_t(x_i) \leftarrow \nabla_{\theta_t} \mathcal{L}(x_i), \quad \tilde{g}_t(x_i) \leftarrow g_t(x_i) \cdot \min(1, C/\|g_t(x_i)\|_2)$
6:     **end for**
7:     $z_t \sim \mathcal{N}(0, \sigma^2 C^2 I_p)$
8:     $\bar{g}_t = \frac{1}{B} \left( \sum_{i=1}^N \tilde{g}_t(x_i) + z_t \right)$
9:     $\theta_{t+1}, m_{t+1}, v_{t+1} \leftarrow \text{AdamUpdate}(\theta_t, m_t, v_t, \bar{g}_t, \beta_1, \beta_2, \gamma)$
10: **end for**
11: **return** $\theta_{TN/B}$

---

**Algorithm 2** AdamUpdate

---

1: **Input:** $\theta_t, m_t, v_t, \bar{g}_t, \beta_1, \beta_2, \gamma$
2: $m_{t+1} \leftarrow \beta_1 \cdot m_t + (1 - \beta_1) \cdot \bar{g}_t, \quad v_{t+1} \leftarrow \beta_2 \cdot v_t + (1 - \beta_2) \cdot \bar{g}_t^2$
3: $\widehat{m}_{t+1} \leftarrow m_{t+1}/(1 - \beta_1^t), \quad \widehat{v}_{t+1} \leftarrow v_{t+1}/(1 - \beta_2^t)$
4: $\theta_{t+1} \leftarrow \theta_t - \alpha \cdot \widehat{m}_{t+1}/\left(\sqrt{\widehat{v}_{t+1}} + \gamma\right)$
5: **return** $\theta_{t+1}, m_{t+1}, v_{t+1}$

---

## B  PRIVACY ACCOUNTING

We train all models under approximate-DP (Dwork et al., 2014), and we view two datasets as being adjacent if and only if one can be obtained from the other by including an extra record (Mironov et al., 2019). Instead of accounting the privacy loss with Moments Accountant (Abadi et al., 2016), we perform computation through $(i)$ Rényi DP (Mironov, 2017; Mironov et al., 2019), $(ii)$ Gaussian DP with an associated central limit theorem (Dong et al., 2019), and $(iii)$ numerically composing tradeoff functions via fast Fourier transform (Gopi et al., 2021; Koskela et al., 2020). All approaches are improvements over the Moments Accountant. Accounting loss with Rényi DP provides strict upper bounds on the actual privacy leakage but may result in loose bounds. Accounting loss with Gaussian DP and its central limit theorem, although asymptotically exact, only provides approximations to the actual loss under a finite number of compositions (Dong et al., 2019, Theorem 3.4). Gopi et al. (2021) observed that accounting loss with GDP and its CLT results in underestimation and proposed to numerically compose tradeoff functions resulting in both upper and lower bounds on the actual leakage $\epsilon$. We therefore also report the converted $\epsilon$ with the approach by Gopi et al. (2021) using their code.[11]

Given the noise multiplier $\sigma$, sampling rate $q$, number of steps $S$, and failure constant $\delta$, $\epsilon$ can be computed via first computing the Rényi DP leakage and then converting it to approximate DP. When a privacy spending (specified by a set of given $\epsilon$ and $\delta$) is prescribed, we can numerically invert the above procedure to obtain a suitable $\sigma$ for noisy optimization. This is what we do throughout all

---

[10]https://github.com/tensorflow/privacy/blob/7c4f5bab0964bd32b7ceafa009d9488920856440/tensorflow_privacy/privacy/optimizers/dp_optimizer.py#L385
[11]https://github.com/microsoft/prv_accountant

experiments. For completeness, given the $\sigma$ chosen as above, we also report the leakage estimated by going through the central limit theorem in Gaussian DP (Bu et al., 2020).

Model selection from hyperparameter tuning on private training and validation data incurs extra leakage (Liu & Talwar, 2019; Chaudhuri & Vinterbo, 2013; Papernot & Steinke, 2021). We perform tuning only on the E2E task and reuse almost the exact hyperparameters for remaining tasks. Note the general strategy of tuning hyperparameters on a separate (public) dataset and thereafter transfer has been applied to train private production language models (Ramaswamy et al., 2020).

## C  ADDITIONAL RESULTS ON MODEL SCALING

We repeat the model scaling experiments from Figure 1 on the other tasks considered in the paper. Figure 5 shows that the trend that larger and better pretrained models lead to improved private fine-tuned performance holds consistently across all tasks. TextHide numbers based on the $\text{TextHide}_{\text{intra}}$ formulation (Huang et al., 2020a).

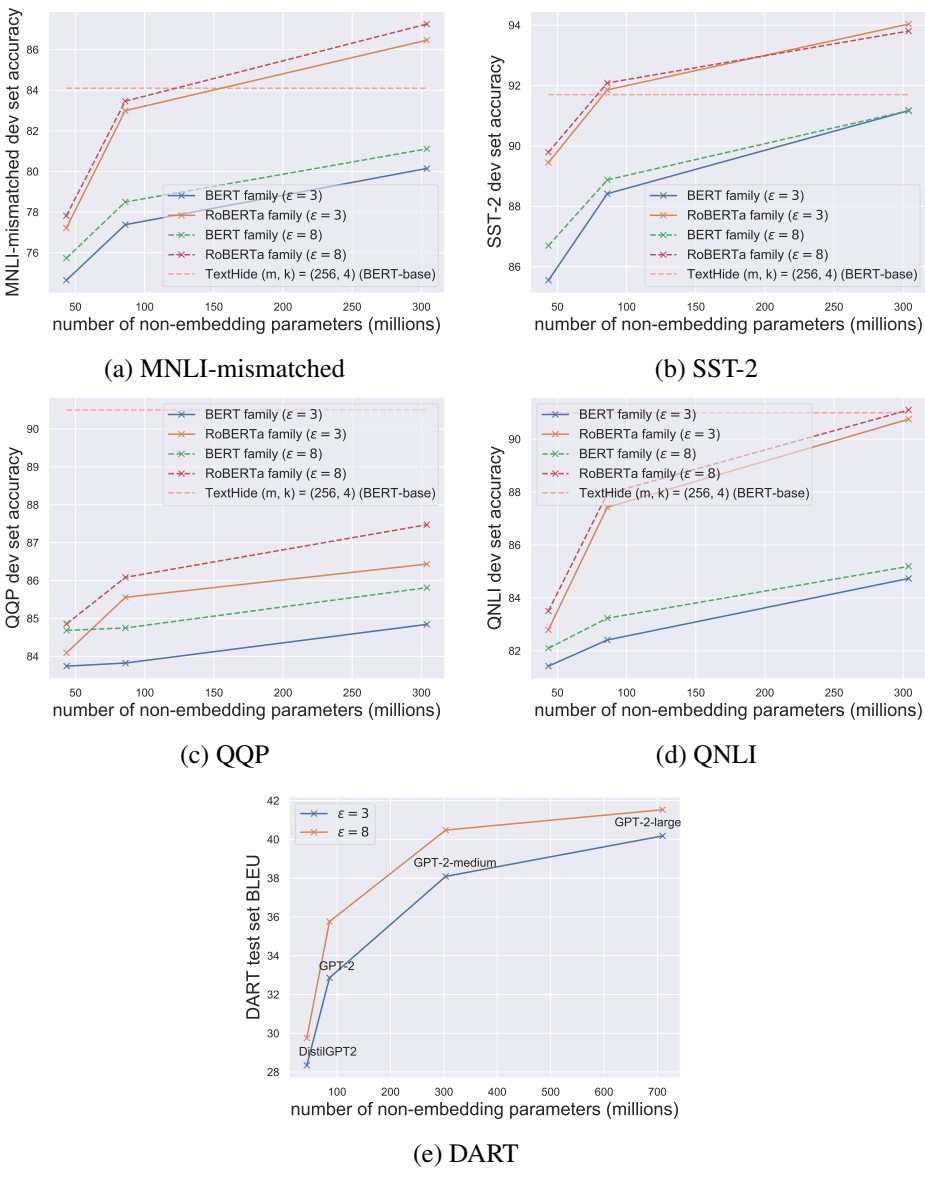

Figure 5: Larger and better pretrained models consistently lead to better private fine-tuned performance on sentence classification and language generation tasks.

## D WHEN AND WHY DOES LINEAR SCALING FAIL?

Recall Tramèr & Boneh (2020) suggested that the following simple rule approximately holds in private learning: Scaling the learning rate together with the batch size by the same constant yields models with almost the same performance. Note that their experiments on MNIST, Fashion-MNIST, and CIFAR-10 used only batch sizes in $\{512, 1024, 2048, 4096\}$. These values are fairly large from a non-private learning perspective. Indeed, our experiments on E2E suggest that this rule does not generalize to batch sizes that are too small (sampling rates $q = {}^B/_N < 2^{-8}$).

We provide an explanation by noting that a core assumption which the linear scaling rule depends on fails to hold for small batch sizes. This assumption is that given a privacy budget, a "square-root" relationship holds between the noise multiplier and the sampling rate (see also (Tramèr & Boneh, 2020, Claim D.1)). For instance, Tramèr & Boneh (2020) showed that $\sigma \approx c\sqrt{q}$ when $q \in [2^{-7}, 1]$ for some constant $c$. Our numerical estimates show that this relationship fails to hold for small $q$ – it underestimates the true noise multiplier $\sigma$ that would be obtained with numerical computation. Figure 6 provides an illustration for $(\epsilon, \delta) = (3, 10^{-5})$ when the sample size $N = 50$k and number of training epochs $E = 50$.

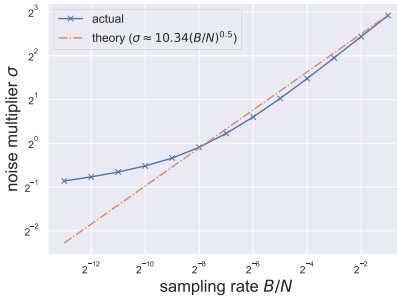

Figure 6: "Square-root" relationship underestimates the noise multiplier for small batch sizes.

## E HOW DOES THE CHOICE OF LABEL WORDS AFFECT SENTENCE CLASSIFICATION

Recall in Section 3.2 we cast sentence classification as filling in the missing word among $K$ candidates for a $K$-way classification problem. Since a label word could be mapped to multiple possible words, we study how the choice of label words affect performance. We again use the sentiment classification task SST-2 as a testbed. Figure 7 shows the effect of varying the label word, where we measure the alignment between the label word and the downstream task by the zero-shot performance of the infilling task (x-axis). We find that increasing the task alignment of the label words alone improves performance by $1 - 2\%$ (orange curve), which is in contrast to the non-private setting, where choice of label words do not affect performance in statistically significant ways (blue curve).

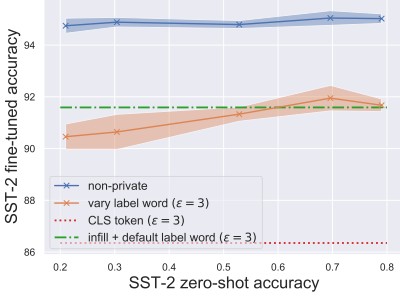

Figure 7: Better text labels help private learning more than non-private learning.

## F    DERIVATION OF THE FROBENIUS NORM IDENTITY

Recall $a \in \mathbb{R}^{B \times T \times d}$ is the input to a linear layer with weight matrix $W \in \mathbb{R}^{p \times d}$, and $g \in \mathbb{R}^{B \times T \times p}$ is the gradient of the loss w.r.t. the output. The identity follows from trivial algebra:

$$
\begin{aligned}
\|\nabla_W \mathcal{L}_i\|_{\mathrm{F}}^2 = \|g_i^\top a_i\|_{\mathrm{F}}^2 &= \left\| \sum_{k=1}^T g_{i,k} a_{i,k}^\top \right\|_{\mathrm{F}}^2 \\
&= \sum_{r=1}^d \sum_{s=1}^p \left( \sum_{k=1}^T a_{i,k,r} g_{i,k,s} \right)^2 \\
&= \sum_{r=1}^d \sum_{s=1}^p \sum_{k_1=1}^T \sum_{k_2=1}^T a_{i,k_1,r} g_{i,k_1,s} a_{i,k_2,r} g_{i,k_2,s} \\
&= \sum_{k_1=1}^T \sum_{k_2=1}^T \left( \sum_{r=1}^d a_{i,k_1,r} a_{i,k_2,r} \right) \left( \sum_{s=1}^p g_{i,k_1,s} g_{i,k_2,s} \right) \\
&= \mathrm{vec}(a_i a_i^\top)^\top \mathrm{vec}(g_i g_i^\top).
\end{aligned}
$$

Note that when $T = 1$, the identity takes the form of

$$
\|\nabla_W \mathcal{L}_i\|_{\mathrm{F}}^2 = \mathrm{vec}(a_i a_i^\top)^\top \mathrm{vec}(g_i g_i^\top) = \|a_i\|_2^2 \|g_i\|_2^2.
$$

This is exactly what is used in the Goodfellow (2015) trick.

## G    PROTOCOL FOR EXPERIMENTS IN SECTION 4.2

For these experiments, we used mock examples with the same format as examples in the E2E dataset. We created mock input sequences of length 100, as this length is almost the maximum length of examples in the actual E2E training set.

Our `JAX` implementation is adapted from a codebase used in the work by Subramani et al. (2020) and utilizes the package `flaxmodels` for loading pretrained models.[12] The `Opacus` (Yousefpour et al., 2021) baseline is based on version `0.14.0` of the library, where for a fair comparison, we also optimized the implementation of the privacy engine by replacing all `einsum` operations with basic primitives that manipulate tensors directly. We found certain `einsum` steps to cause large and unnecessary speed and memory overheads.

To identify the maximum batch size that each approach could use, we ran binary search over the range of possible batch sizes until the upper bound matched the lower bound and that OOM did not occur. To estimate the throughput of each private method, we compared them to non-private training first. Specifically, for a pairwise comparison, we took the maximum batch size for non-private training and some private method, and computed the least common multiple as a batch size to perform updates with. Using the least common multiple batch size ensures that both methods will process exactly the same number of examples and perform the same number of updates. By estimating the time elapse of these methods for performing a fixed number of updates, we obtained throughputs for non-private training and the private method. This gives us the relative throughput of the private method with non-private training as the reference.

For methods implemented in `PyTorch`, the time elapse was recorded with `torch.profiler`. When estimating the time elapse for a training procedure, we first performed 3 gradients updates as a warm up process before taking the actual steps which will be timed. In particular, this eliminates the time that `JAX` uses for compiling computation graphs with `vmap` and `jit`.

All experimental runs in the section were under full precision.

---

[12] https://github.com/matthias-wright/flaxmodels

## H DETAILS AND ADDITIONAL RESULTS FOR STUDIES IN SECTION 3.1

Table 4: Default hyperparameters for ablation studies.

| Method | Full |
|---|---|
| DP guarantee $(\epsilon, \delta)$ | $(3, 1/2|\mathcal{D}_{\text{train}}|)$ |
| Clipping norm $C$ | 0.1 |
| Batch size $B$ | 1024 |
| Learning rate $\eta$ | $10^{-3}$ |
| Learning rate decay | no |
| Epochs $E$ | 10 for E2E; 3 for SST-2 |
| Weight decay $\lambda$ | 0 |
| Noise scale $\sigma$ | calculated numerically so that a DP budget of $(\epsilon, \delta)$ is spent after $E$ epochs |

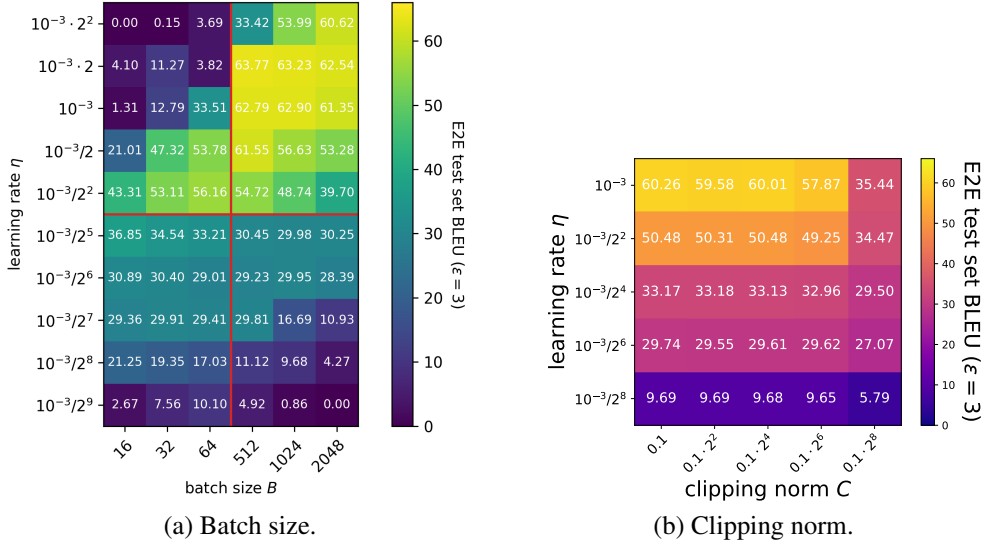

(a) Batch size.                    (b) Clipping norm.

Figure 8: Additional results on hyperparameter sensitivity.

## I HYPERPARAMETER SEARCH RANGES FOR EXPERIMENTS IN SECTION 5

We compare different adaptation methods by reporting task specific metrics on the test split using hyperparameters that maximize validation BLEU on E2E. For sentence classification tasks, we reused the same hyperparameters, except for the number of training epochs and batch size. For SST-2, we reused the same batch size as for private E2E fine-tuning, and the number of epochs exactly as in typical non-private fine-tuning for SST-2 (number of epochs equals to 3 in this case). For remaining classification tasks, we use a batch size such that the sampling rate is the same as for SST-2, and a number of training epochs that is roughly proportional to the dataset size. Appendix L outlines why we transfer the sampling rate as opposed to the batch size. We list the range of hyperparameters that we searched over for each individual adaptation method on E2E considered in the paper. Prefix-tuning has two additional hyperparameters: the length of the prefix and the dimensionality of the hidden layer. We set these to the default used by Li & Liang (2021) (10 for the former and 512 for the latter). For Adam, we use the default hyperparamaters set by PyTorch (Paszke et al., 2017).

Table 5: Hyperparameter search range for different methods.

| Method | Full | Prefix | Linear | FT2 |
|---|---|---|---|---|
| Guarantee $(\epsilon, \delta)$ | $(3, 1/2|\mathcal{D}_{\text{train}}|)$ | $(3, 1/2|\mathcal{D}_{\text{train}}|)$ | $(3, 1/2|\mathcal{D}_{\text{train}}|)$ | $(3, 1/2|\mathcal{D}_{\text{train}}|)$ |
| Clipping norm $C$ | 0.1 | 0.1 | 0.1 | 0.1 |
| Batch size $B$ | $\{512, 1024\}$ | $\{512, 1024\}$ | $\{512, 1024\}$ | $\{512, 1024\}$ |
| Learning rate $\eta$ | $\{200, 100, 30, 10, 3\} \cdot 10^{-5}$ | $\{200, 100, 30, 10, 3\} \cdot 10^{-5}$ | $\{200, 100, 30, 10, 3\} \cdot 10^{-5}$ | $\{200, 100, 30, 10, 3\} \cdot 10^{-5}$ |
| LR decay | $\{\text{yes}, \text{no}\}$ | $\{\text{yes}, \text{no}\}$ | $\{\text{yes}, \text{no}\}$ | $\{\text{yes}, \text{no}\}$ |
| Epochs $E$ | $\{10, 30, 50\}$ | $\{10, 30, 50\}$ | $\{10, 30, 50\}$ | $\{10, 30, 50\}$ |
| Weight decay $\lambda$ | 0 | 0 | 0 | 0 |
| Noise scale $\sigma$ | calculated numerically so that a DP budget of $(\epsilon, \delta)$ is spent after $E$ epochs | | | |

Table 6: Hyperparameter search range for different methods (continued).

| Method | LoRA | RGP |
|---|---|---|
| DP guarantee $(\epsilon, \delta)$ | $(3, 1/2|\mathcal{D}_{\text{train}}|)$ | $(3, 1/2|\mathcal{D}_{\text{train}}|)$ |
| Clipping norm $C$ | 0.1 | $\{0.1, 1, 10\}$ |
| Batch size $B$ | $\{512, 1024\}$ | $\{512, 1024\}$ |
| Learning rate $\eta$ | $\{300, 100, 30, 10, 3\} \cdot 10^{-5}$ | $\{300, 100, 30, 10, 3\} \cdot 10^{-5}$ |
| LR decay | $\{\text{yes}, \text{no}\}$ | $\{\text{yes}, \text{no}\}$ |
| Epochs $E$ | $\{5, 10, 30, 50\}$ | $\{5, 10, 30, 50\}$ |
| Weight decay $\lambda$ | 0 | 0 |
| Rank $k$ | $\{1, 2, 4, 8\}$ | $\{1, 2, 4, 8\}$ |
| Noise scale $\sigma$ | calculated numerically so that a DP budget of $(\epsilon, \delta)$ is spent after $E$ epochs | |

## J  FULL RESULTS FOR EXPERIMENTS IN SECTION 5.2

Table 7: Full results on E2E from fine-tuning GPT-2.

| Method | DP Guarantee | Gaussian DP +CLT | Compose tradeoff func. | BLEU | NIST | METEOR | ROUGE-L | CIDEr |
|---|---|---|---|---|---|---|---|---|
| full | $\epsilon = 3$ | $\epsilon \approx 2.33$ | $\epsilon \approx 2.67$ | 61.519 | 6.697 | 0.384 | 65.670 | 1.761 |
| | $\epsilon = 8$ | $\epsilon \approx 5.51$ | $\epsilon \approx 6.98$ | 63.189 | 7.444 | 0.400 | 66.429 | 1.919 |
| | non-private | - | - | 69.463 | 8.780 | 0.461 | 71.359 | 2.422 |
| LoRA | $\epsilon = 3$ | $\epsilon \approx 2.68$ | $\epsilon \approx 2.75$ | 58.153 | 5.463 | 0.370 | 65.773 | 1.581 |
| | $\epsilon = 8$ | $\epsilon \approx 6.77$ | $\epsilon \approx 7.28$ | 63.389 | 7.449 | 0.407 | 67.525 | 1.948 |
| | non-private | - | - | 69.682 | 8.822 | 0.463 | 71.709 | 2.491 |
| prefix | $\epsilon = 3$ | $\epsilon \approx 2.33$ | $\epsilon \approx 2.67$ | 47.772 | 5.775 | 0.331 | 58.964 | 1.300 |
| | $\epsilon = 8$ | $\epsilon \approx 5.51$ | $\epsilon \approx 6.98$ | 49.263 | 6.276 | 0.349 | 60.730 | 1.496 |
| | non-private | - | - | 68.845 | 8.722 | 0.456 | 70.805 | 2.418 |
| RGP | $\epsilon = 3$ | $\epsilon \approx 2.18$ | $\epsilon \approx 2.59$ | 58.482 | 5.249 | 0.363 | 65.560 | 1.507 |
| | $\epsilon = 8$ | $\epsilon \approx 5.19$ | $\epsilon \approx 6.89$ | 58.455 | 5.525 | 0.364 | 65.030 | 1.569 |
| | non-private | - | - | 68.328 | 8.722 | 0.445 | 68.844 | 2.345 |
| top2 | $\epsilon = 3$ | $\epsilon \approx 2.68$ | $\epsilon \approx 2.75$ | 25.920 | 1.510 | 0.197 | 44.536 | 0.452 |
| | $\epsilon = 8$ | $\epsilon \approx 6.77$ | $\epsilon \approx 7.28$ | 26.885 | 1.547 | 0.207 | 46.421 | 0.499 |
| | non-private | - | - | 65.752 | 8.418 | 0.443 | 68.704 | 2.180 |
| retrain | $\epsilon = 3$ | $\epsilon \approx 2.33$ | $\epsilon \approx 2.67$ | 15.457 | 0.376 | 0.113 | 35.240 | 0.116 |
| | $\epsilon = 8$ | $\epsilon \approx 5.51$ | $\epsilon \approx 6.98$ | 24.247 | 1.010 | 0.145 | 39.951 | 0.281 |
| | non-private | - | - | 65.731 | 8.286 | 0.429 | 68.751 | 2.004 |

Table 8: Full results on DART from fine-tuning GPT-2. Trend is consistent with results on E2E.

| Method | DP Guarantee | Gaussian DP + CLT | Compose tradeoff func. | METEOR | ROUGE-1 | ROUGE-2 | ROUGE-L | BLEU | BERTScore | BLEURT |
|---|---|---|---|---|---|---|---|---|---|---|
| full | $\epsilon = 3$ | $\epsilon \approx 2.28$ | $\epsilon \approx 2.65$ | 0.294 | 62.815 | 40.773 | 52.063 | 31.025 | 0.887 | -0.058 |
| | $\epsilon = 8$ | $\epsilon \approx 5.35$ | $\epsilon \approx 6.95$ | 0.319 | 66.423 | 43.609 | 54.576 | 35.057 | 0.901 | 0.043 |
| | non-private | - | - | 0.369 | 71.563 | 47.168 | 56.717 | 42.783 | 0.915 | 0.178 |
| LoRA | $\epsilon = 3$ | $\epsilon \approx 2.68$ | $\epsilon \approx 2.76$ | 0.304 | 63.641 | 40.753 | 52.012 | 32.329 | 0.885 | -0.029 |
| | $\epsilon = 8$ | $\epsilon \approx 6.68$ | $\epsilon \approx 7.26$ | 0.318 | 66.336 | 43.056 | 54.082 | 34.163 | 0.899 | 0.036 |
| | non-private | - | - | 0.366 | 71.192 | 47.336 | 57.430 | 42.254 | 0.915 | 0.182 |
| prefix | $\epsilon = 3$ | $\epsilon \approx 2.28$ | $\epsilon \approx 2.65$ | 0.269 | 59.503 | 38.229 | 49.444 | 25.726 | 0.860 | -0.144 |
| | $\epsilon = 8$ | $\epsilon \approx 5.35$ | $\epsilon \approx 6.95$ | 0.297 | 64.009 | 41.581 | 52.602 | 30.463 | 0.892 | -0.021 |
| | non-private | - | - | 0.353 | 70.341 | 46.643 | 56.858 | 40.163 | 0.912 | 0.148 |
| RGP | $\epsilon = 3$ | $\epsilon \approx 2.10$ | $\epsilon \approx 2.54$ | 0.265 | 58.688 | 37.202 | 49.011 | 25.748 | 0.873 | -0.175 |
| | $\epsilon = 8$ | $\epsilon \approx 5.02$ | $\epsilon \approx 6.86$ | 0.279 | 60.005 | 38.258 | 49.835 | 28.304 | 0.874 | -0.141 |
| | non-private | - | - | 0.324 | 65.667 | 42.617 | 53.477 | 35.551 | 0.895 | 0.022 |
| top2 | $\epsilon = 3$ | $\epsilon \approx 2.68$ | $\epsilon \approx 2.76$ | 0.022 | 3.570 | 2.183 | 3.166 | 0.388 | 0.098 | -1.952 |
| | $\epsilon = 8$ | $\epsilon \approx 6.68$ | $\epsilon \approx 7.26$ | 0.054 | 11.475 | 7.054 | 10.042 | 2.453 | 0.240 | -1.660 |
| | non-private | - | - | 0.318 | 62.777 | 38.367 | 49.426 | 36.099 | 0.883 | -0.082 |
| retrain | $\epsilon = 3$ | $\epsilon \approx 2.28$ | $\epsilon \approx 2.65$ | 0.064 | 19.085 | 8.901 | 17.142 | 2.997 | 0.493 | -1.513 |
| | $\epsilon = 8$ | $\epsilon \approx 5.35$ | $\epsilon \approx 6.95$ | 0.093 | 24.971 | 11.938 | 21.680 | 7.765 | 0.573 | -1.302 |
| | non-private | - | - | 0.232 | 47.782 | 26.361 | 37.864 | 26.794 | 0.806 | -0.593 |

## K    Details for experiments in section 5.2

**Sentence Classification.**    Results for RGP in Table 1 are taken from documented numbers in their released codebase.[13]  These results are under the DP guarantees of $(\epsilon, \delta) = (3, 10^{-5})$ or $(\epsilon, \delta) = (8, 10^{-5})$. These guarantees are strictly looser than our guarantees which are based on $\delta = 1/2|\mathcal{D}_{\text{train}}|$ (recall the smallest dataset in this cohort of tasks has 60k+ records). The RGP numbers in Table 1 are higher than those reported in their paper (Yu et al., 2021c), since the latter numbers are not based on fine-tuning the official RoBERTa models.

**Table-To-Text Generation.**    To evaluate models trained on E2E and DART, we evaluate generations from models obtained with beam search with a beam size of 5. For evaluation, we run the official pipeline for E2E,[14] and the pipeline used in the GEM benchmark (Gehrmann et al., 2021) for DART.[15]

**Chit-Chat Dialog Generation.**    We built off Huggingface's codebase of the winning entry of the ConvAI2 competition, [16] [17] and used their preprocessed training set with the minor modification of truncating the number of training examples to be a multiple of the batch size. The original ConvAI2 competition is aimed at advancing research on building engaging chatbots and also requested models to predict the mostly likely response given a list of candidates. The challenge included hits@1 as part of its suite of automatic metrics. For simplicity, we skip this step of predicting the most likely response for both training and evaluation. Results for the entry HuggingFace (ConvAI2 winner) in Table 3 are taken from the official validation set leader board.[18] Our reimplementation of HuggingFace's submission uses the released code for their winning entry, fine-tunes GPT with the default hyperparameters, and removes the classification loss for learning to predict the most likely response given candidates.

We additionally fine-tuned DialoGPT-medium (Zhang et al., 2019), since the model was pretrained on conversation-like exchanges extracted from Reddit comment chains. Intuitively, this pretraining corpus is more aligned with the downstream fine-tuning data than WebText (Radford et al., 2019), and thus would likely improve downstream performance.

To evaluate the F1 score, we obtained predicted responses from trained models using beam search with a beam size of 5. Since past work found that the F1 score can be gamed by always letting the model predict a predetermined phrase (Dinan et al., 2019), we additionally ask humans to rate generations from the model through Amazon mechanical turk to obtain a more complete picture of model quality. When sampling responses for human evaluation, we used nucleus sampling with $p = 0.9$ (Holtzman et al., 2019). For human evaluation, we asked 20 turkers to each rate 5 entries. For each entry, a turker is asked to rate on a scale of 1 to 5 the quality of predicted responses from privately fine-tuned DialoGPT-medium models, non-privately fine-tuned DialoGPT-medium models, non-privately fine-tuned GPT models (our reimplementation of HuggingFace's entry), and the reference text, given the history of the dialog. Since human evaluation can yield noisy results, we also report the 95% asymptotic confidence interval in Table 3. All models were trained and evaluated on the version of Persona-Chat with the original persona. All numbers reported in Table 3 are obtained on the validation split.

## L    Transferring Hyperparameters Across Datasets

Our work involved tuning hyperparameter with models trained via DP-Adam on one private dataset and transferring such hyperparameters to other private datasets. Since different datasets may be of different sizes, transferring the batch size may cause a discrepancy in the effective noise multiplier across workloads with different datasets. Transferring the batch size based on hyperparameter tuning on small datasets to larger datasets can be particularly problematic, as the effective noise multiplier

---

[13]https://github.com/dayu11/Differentially-Private-Deep-Learning/tree/main/language
[14]https://github.com/tuetschek/e2e-metrics
[15]https://github.com/GEM-benchmark/GEM-metrics
[16]https://github.com/huggingface/transfer-learning-conv-ai
[17]http://convai.io/2018/
[18]https://github.com/DeepPavlov/convai/blob/master/leaderboards.md

## M DOES DP FINE-TUNING PREVENT UNINTENDED MEMORIZATION?

One of the ultimate goals of fitting models under DP is to ensure that training data extraction is unlikely given the trained model. To empirically evaluate whether DP fine-tuning helps prevent against unintended memorization and related attacks, we follow the secret sharer framework (Carlini et al., 2019) and estimate the *exposure* of artificial canaries inserted into the training set used for fine-tuning. We use the E2E dataset as a testbed.

To create canaries, we first form a subvocabulary by randomly sampling $V = 10$ words in the original vocabulary of GPT-2. Our canaries have prefixes of the form

```
" name :    <word> | Type :    <word> | area :    <word>",
```

where `<word>` is randomly sampled from the subvocabulary. The suffix which our model should learn to predict consists of randomly sampled words with an average length of $l = 5$. By definition, canaries with an estimated exposure close to $\log_2(V^l) \approx 17$ can likely be extracted. We experiment with canary-corrupted datasets for repetition values $r \in \{1, 10, 100\}$. A canary has a higher chance in being extracted when it's repeated for more than once in the training data.

Table 9: Fine-tuning under DP prevents unintended memorization of downstream data. Numbers reported are exposure values estimated with the *approximation by distribution model* approach.

| Guarantee \ Repetitions | $r = 1$ | $r = 10$ | $r = 100$ |
|---|---|---|---|
| $\epsilon = 3$ | $1.09 \pm 0.86$ | $1.32 \pm 1.32$ | $5.26 \pm 4.20$ |
| non-private | $13.82 \pm 3.86$ | $17.22 \pm 0.00$ | $17.78 \pm 5.49$ |

## N TEMPLATES AND LABEL WORDS FOR TEXT-INFILLING-BASED CLASSIFICATION IN SECTION 3.2

Recall that fine-tuning for classification can be reformulated as filling in the `[MASK]` token in a template sequence. Here, we list the templates used for each classification task considered in the paper. These templates are almost generic and are not obtained from expensive manual or automated search. We anticipate better templates obtained from automated search based on data (Gao et al., 2020) to improve the performance even further. However, we also expect that such a procedure would lead to some amount of increased privacy spending if it were based on private data.

| Task | Template | Label words |
|---|---|---|
| SST-2 | $<S_1>$ It was [MASK] . | positive: great, negative: terrible |
| MNLI | $<S_1>$ ? [MASK] , $<S_2>$ | entailment: Yes, netural: Maybe, contradiction: No |
| QNLI | $<S_1>$ ? [MASK] , $<S_2>$ | entailment: Yes, not_entailment: No |
| QQP | $<S_1>$ [MASK] , $<S_2>$ | equivalent: Yes, not_equivalent: No |

Table 10: Templates and label words borrowed from the work by Gao et al. (2020).

## O  UNCURATED SAMPLES FROM FINE-TUNED MODELS

| | |
|---|---|
| Table | name : The Punter — Type : restaurant — food : Indian — price : cheap — customer rating : average — area : riverside — family friendly : no — near : Express by Holiday Inn |
| GPT-2 ($\epsilon = 3$) | The Punter is a cheap Indian restaurant near Express by Holiday Inn in the riverside area. It is not family - friendly. |
| GPT-2 ($\epsilon = 8$) | The Punter is a cheap Indian restaurant near Express by Holiday Inn in the riverside area. It is not family - friendly. |
| GPT-2-m ($\epsilon = 3$) | The Punter is a cheap Indian restaurant located in the riverside area near Express by Holiday Inn. It has an average customer rating and is not family - friendly. |
| GPT-2-m ($\epsilon = 8$) | The Punter is a restaurant providing Indian food in the cheap price range. It is located in the riverside area near Express by Holiday Inn. Its customer rating is average. |
| GPT-2-l ($\epsilon = 3$) | The Punter is a cheap Indian restaurant in the riverside area near Express by Holiday Inn. It is not family - friendly and has an average customer rating. |
| GPT-2-l ($\epsilon = 8$) | The Punter is a restaurant providing Indian food in the cheap price range. It is located in the riverside area near Express by Holiday Inn. Its customer rating is average. |
| Reference | The restaurant named The Punter has cheap Indian food and an average customer rating . It is near the Express by Holiday Inn on the riverside and is not family friendly . |
| Table | name : The Mill — Type : restaurant — food : English — price : moderate — customer rating : 3 out of 5 — area : city centre — family friendly : yes — near : Café Rouge |
| GPT-2 ($\epsilon = 3$) | The Mill is a moderately priced restaurant located in the city centre near Café Rouge. |
| GPT-2 ($\epsilon = 8$) | The Mill is a moderately priced restaurant located in the city centre near Café Rouge. |
| GPT-2-m ($\epsilon = 3$) | The Mill is an English restaurant located in the city centre near Café Rouge. It is moderately priced and has a customer rating of 3 out of 5. |
| GPT-2-m ($\epsilon = 8$) | The Mill is a moderately priced restaurant located in the city centre near Café Rouge. It is child friendly and has a customer rating of 3 out of 5. |
| GPT-2-l ($\epsilon = 3$) | The Mill is a moderately priced English restaurant in the city centre near Café Rouge. It is child friendly and has a customer rating of 3 out of 5. |
| GPT-2-l ($\epsilon = 8$) | The Mill is a kid friendly English restaurant in the city centre near Café Rouge. It has a moderate price range and a customer rating of 3 out of 5. |
| Reference | Serving moderately priced English food with a 3 out of 5 customer approval , The Mill restaurant is kid friendly and conveniently located at the city centre near the Café Rouge . |
| Table | name : The Vaults — Type : pub — food : Japanese — price : high — customer rating : 3 out of 5 — area : city centre — family friendly : yes — near : Raja Indian Cuisine |
| GPT-2 ($\epsilon = 3$) | Located near Raja Indian Cuisine in the city centre, The Vaults is a family - friendly Japanese pub with a high customer rating. |
| GPT-2 ($\epsilon = 8$) | The Vaults is a Japanese pub in the city centre near Raja Indian Cuisine. It has a high price range and a customer rating of 3 out of 5. |
| GPT-2-m ($\epsilon = 3$) | The Vaults is a Japanese pub located in the city centre near Raja Indian Cuisine. It has a high price range and a customer rating of 3 out of 5. |
| GPT-2-m ($\epsilon = 8$) | The Vaults is a Japanese pub located in the city centre near Raja Indian Cuisine. It has a high price range and a customer rating of 3 out of 5. |
| GPT-2-l ($\epsilon = 3$) | The Vaults is a Japanese pub in the city centre near Raja Indian Cuisine. It has a high price range and a customer rating of 3 out of 5. |
| GPT-2-l ($\epsilon = 8$) | The Vaults is a child friendly Japanese pub in the city centre near Raja Indian Cuisine. It has a high price range and a customer rating of 3 out of 5. |
| Reference | Located near the Raja Indian Cuisine at the city centre , is family - friendly pub The Vaults . Serving Japanese food , The Vaults , is high priced with a 3 out of 5 customer satisfaction rating . |

Table 11: Fully fine-tuned GPT-2, GPT-2-medium, and GPT-2-large generations with E2E test table entries.

| Table | Real Madrid Castilla : manager : Luis Miguel Ramis — Abner (footballer) : club : Real Madrid Castilla — Abner (footballer) : club : C.D. FAS |
|---|---|
| GPT-2 ($\epsilon = 3$) | Luis Miguel Ramis played for Real Madrid Castilla and played for C.D. FAS. |
| GPT-2 ($\epsilon = 8$) | Luis Miguel Ramis is the manager of Abner (footballer) who plays for Real Madrid Castilla. |
| GPT-2-m ($\epsilon = 3$) | Luis Miguel Ramis is the manager of Real Madrid Castilla. He plays for C.D. FAS. |
| GPT-2-m ($\epsilon = 8$) | Luis Miguel Ramis is the manager of Real Madrid Castilla. He plays for C.D. FAS. |
| GPT-2-l ($\epsilon = 3$) | Luis Miguel Ramis is the manager of Real Madrid Castilla and C.D. FAS. |
| GPT-2-l ($\epsilon = 8$) | Luis Miguel Ramis is the manager of Real Madrid Castilla and Abner (footballer) plays for C.D. FAS. |
| Reference | Footballer, Abner, plays C.D. FAS. and Real Madrid Castilla, the manager of which, is Luis Miguel Ramis. |

| Table | United States : ethnic_group : Asian Americans — United States : capital : Washington, D.C. — Albany, Oregon : is_part_of : Benton County, Oregon — Albany, Oregon : country : United States |
|---|---|
| GPT-2 ($\epsilon = 3$) | The capital of the United States is Washington, D.C. and is part of Benton County, Oregon. |
| GPT-2 ($\epsilon = 8$) | The capital of the United States is Washington, D.C. and is part of Benton County, Oregon. |
| GPT-2-m ($\epsilon = 3$) | Albany, Oregon is part of Benton County, Oregon in the United States where Asian Americans are an ethnic group. |
| GPT-2-m ($\epsilon = 8$) | Albany, Oregon is part of Benton County, Oregon in the United States where Asian Americans are an ethnic group. |
| GPT-2-l ($\epsilon = 3$) | Albany, Oregon is part of the United States where Asian Americans are an ethnic group and the capital is Washington D.C. |
| GPT-2-l ($\epsilon = 8$) | Albany, Oregon is part of the United States where Asian Americans are an ethnic group and the capital is Washington D.C. |
| Reference | The Asian Americans are an ethnic group in the United States, which has the capital city of Washington DC. It is also the location of Albany, part of Benton County in Oregon. |

| Table | A Loyal Character Dancer : language : English language — English language : spoken_in : Great Britain — A Loyal Character Dancer : country : United States — United States : ethnic_group : Native Americans in the United States |
|---|---|
| GPT-2 ($\epsilon = 3$) | A Loyal Character Dancer is an English language spoken in the United States where Native Americans are the ethnic group. |
| GPT-2 ($\epsilon = 8$) | A Loyal Character Dancer is written in English and is spoken in Great Britain. Native Americans are an ethnic group in the United States. |
| GPT-2-m ($\epsilon = 3$) | A Loyal Character Dancer is written in English and is written in the United States where Native Americans are an ethnic group. |
| GPT-2-m ($\epsilon = 8$) | A Loyal Character Dancer is written in English and is written in the United States where Native Americans are an ethnic group. |
| GPT-2-l ($\epsilon = 3$) | A Loyal Character Dancer is written in English and is written in the United States where Native Americans are an ethnic group. |
| GPT-2-l ($\epsilon = 8$) | A Loyal Character Dancer is written in English, which is spoken in Great Britain and the United States. Native Americans are an ethnic group in the United States. |
| Reference | A Loyal Character Dancer is written in English which is spoken in Great Britain. The book is published in the United States where the Native Americans are an ethnic group. |

Table 12: Fully fine-tuned GPT-2, GPT-2-medium, and GPT-2-large generations with DART test table entries.

| Persona | ['i am 40 years old.', 'i work as a car salesman.', 'my wife spends all my money.', 'i hate my job.', "i'm planning on getting a divorce."] |
|---|---|
| History | ['i know the struggle, i volunteer at a homeless shelter', "great, i hope you don't get to see me there.", 'i hope not either, would not wish it on my worse enemy', 'between wife, job with no money, i am thinking now of divorce.', "i've given it a few thoughts before"] |
| GPT-2-m ($\epsilon = 3$) | i wish i had a better idea . |
| GPT-2-m ($\epsilon = 8$) | i think i will . what are your thoughts on divorce ? |
| DialoGPT-medium ($\epsilon = 3$) | i'm sorry to hear that . i hope you don't have any financial problems . |
| DialoGPT-medium ($\epsilon = 8$) | i don't have any money , i do not know how i could manage . |
| Reference | do you give demonstrations of your skills ? |

| Persona | ['my mother is a dentist.', 'my favourite artist is travis scott.', 'i have one older sister.', 'i am currently looking for a job preferably at a restaurant.', 'i like playing basketball.'] |
|---|---|
| History | ['hello, how are you tonight?'] |
| GPT-2-m ($\epsilon = 3$) | i'm good , good luck |
| GPT-2-m ($\epsilon = 8$) | i am good , just listening to travis scott . what about you ? |
| DialoGPT-medium ($\epsilon = 3$) | i'm good . just got back from a dentist visit . |
| DialoGPT-medium ($\epsilon = 8$) | i am doing well . just got back from a dentist visit |
| Reference | i am good . about to go to basketball practice . |

| Persona | ['my mom passed away when i was young.', 'i work in the insurance business.', 'my father is in the army.', 'i recently was fired and had to get a new job.'] |
|---|---|
| History | ["god isn't really real", 'to each their own, but i believe he is.', 'my children are gone now', 'i lost my mom at a young age', 'my wife took them all'] |
| GPT-2-m ($\epsilon = 3$) | she was a great mother |
| GPT-2-m ($\epsilon = 8$) | they were so young , they were so crazy |
| DialoGPT-medium ($\epsilon = 3$) | that is horrible . my mom was a soldier |
| DialoGPT-medium ($\epsilon = 8$) | what happened to her ? |
| Reference | sorry to hear that . i barely see my dad he's in the army |

| Persona | ['i am also a graduate student.', 'i walk dogs for a living.', 'my parents taught me survival skills.', 'i am an avid kayaker.', 'i enjoy reading journals and guides related to psychology.'] |
|---|---|
| History | ['hi, how are you today?', "good evening. how are you tonight? i'm well.", 'i am doing good, i just finished playing ping pong.', 'i am trying to write invoices for my dog walking business.', 'that seems like hare work, i like to play the violin in my free time.'] |
| GPT-2-m ($\epsilon = 3$) | i have a lot of dogs , i walk them on my walks . |
| GPT-2-m ($\epsilon = 8$) | that is a very rewarding hobby . i have a dog named jasper . |
| DialoGPT-medium ($\epsilon = 3$) | oh , that is a nice hobby . what do you do for a living ? |
| DialoGPT-medium ($\epsilon = 8$) | it is . i love learning to play the violin . |
| Reference | i love the violin . i do not play anything . i kayak for fun however . |

Table 13: Fully fine-tuned GPT-2-medium and DialoGPT-medium generations for Persona-Chat validation examples.

## P    ADDITIONAL RELATED WORK

**Differentially Private Deep Learning.**    DP-SGD has been viewed as ineffective for large models due to the addition of large Gaussian noise to gradient updates. Improvements to the learning procedure mostly fall under two distinct camps: $(i)$ Simplifying the private learning problem, and $(ii)$ reducing the scale of noise. For instance, Papernot et al. (2019); Tramèr & Boneh (2020); Abadi et al. (2016) consider transferring features learned on public datasets to simplify the subsequent private learning task. On the other hand, Zhou et al. (2020); Kairouz et al. (2020) remove the ambient dimension dependence of DP noise by identifying subspaces in which private gradients lie and would be privatized. Yu et al. (2021b;c) make such ideas practical and demonstrate improved results on private learning benchmarks. Zhang et al. (2021) applied the sparse vector technique to learning wide neural layers to reduce the amount of injected noise. Our work mostly falls under the first camp – improving private learning through simplifying the learning task. Our work is also distinct from prior works in that we focus on privately fine-tuning large pretrained models. Lastly, there are alternative solutions in the literature that enforces DP which are not based on gradient perturbation (Papernot et al., 2018; 2016). These methods typically require extra public data and are not the present focus.

**Parameter-Efficient Fine-Tuning.**    Recent developments on pretrained model adaptation have produced a wide range of parameter-efficient fine-tuning methods for both vision and language tasks. We briefly summarize these, grouping by category. Approaches based on optimizing prompt-like constructions for NLP tasks include prefix-tuning (Li & Liang, 2021), P-tuning (Liu et al., 2021), and prompt-tuning (Lester et al., 2021). Adapter-based methods insert small subnetworks inside pretrained Transformers (Houlsby et al., 2019; Rücklé et al., 2020; Pfeiffer et al., 2020). Methods that optimize low-rank matrices include the work by Hu et al. (2021); Mahabadi et al. (2021). In addition, there are adaptation methods that only optimize biases for vision (Cai et al., 2020) and language tasks (Ben Zaken et al., 2021). Our evaluation in Section 5.2 covered the most representative methods that generally have state-of-the-art non-private learning performance (at the time of writing) for the range of NLP tasks studied in this paper.

**Speeding Up DP-SGD.**    Apart from the work by Lee & Kifer (2020) and Subramani et al. (2020), there is an approach that approximates per-example gradient norms through the combination of random projection and forward-mode autodiff (Bu et al., 2021). While faster than vanilla private learning, this approach has the drawback of increased privacy spending and having an extra hyperparameter. Our ghost clipping technique, while only suited for Transformers applied to sequential data, does not introduce new hyperparameters.

**Alternative Clipping Strategies.**    While there are alternative clipping strategies in the literature that show improvements on simple tasks (Pichapati et al., 2019; Asi et al., 2021), we have opted to study the simplest strategy that clips gradients by their Euclidean norm. We leave the study of these algorithms for NLP tasks to future work.

**Concurrent Work.**    We are made aware of a concurrent work that also studies fine-tuning large language models under DP (Yu et al., 2021a). This work presents initial successes on fine-tuning under DP with low-rank methods such as LoRA. Our experiments on language generation (see Section 5.2 and Table 2) demonstrate similar findings. Yet, we moreover show that full fine-tuning with good hyperparameters attains similar performance and possesses similar model scaling properties, which was raise by Yu et al. (2021a) as interesting open questions to pursue. Lastly, our private fine-tuning results for sentence classification may be far from optimal, since we used hyperparameters mostly transferred from tuning on the E2E language generation task.

**DP Synthetic Data Generation.**    Fine-tuning generative language models on private data under DP can also be viewed as a means of accomplishing DP synthetic data generation – learning generative models from private data so that synthetic examples could be sampled and used for analysis. Previous work employed generative adversarial networks and focused primarily on image or tabular datasets (Torkzadehmahani et al., 2019; Neunhoeffer et al., 2020; Chen et al., 2020; Torfi et al., 2020). Perhaps more related is the work by Bommasani et al. (2019) which attempted fine-tuning GPT-2 on medical datasets to generate synthetic records but did not report any quantitative results.

## Q    ESTIMATES OF RUN-TIME IN PRACTICE

The actual run-time of algorithms depends on implementation details. Here, we outline estimates of the run-time for full fine-tuning with DP-Adam on tasks considered in the paper. These numbers are based on running with a single RTX 3090 with `PyTorch==1.9.0`. Fine-tuning GPT-2 on E2E and DART takes less than 10 minutes per epoch, and fine-tuning for 10 epochs results in reasonably performing models. The time to fine-tune RoBERTa-base on classification tasks depends on the size of the dataset. It takes less than 10 minutes per epoch on the smallest SST-2, whereas for the largest MNLI, it takes less than an hour per epoch.

## R    EFFECTS OF VARYING $\epsilon$ AND $\delta$ ON DP FULL FINE-TUNING PERFORMANCE

Figure 9 shows how different values of $\epsilon$ and $\delta$ affect the performance of full fine-tuning under DP.

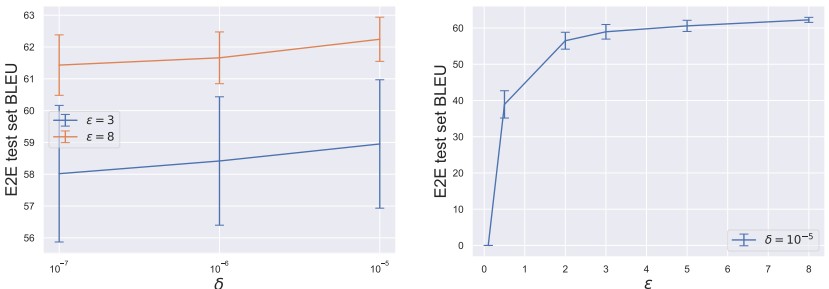

Figure 9: **Left:** $\delta$ affects performance marginally. **Right:** $\epsilon$ affects performance more significantly when $\epsilon < 2$. Errorbars are one standard deviation away from the mean over five independent runs.

## S    DP-ADAM VS DP-SGD

Our work focused on DP-Adam as opposed to DP-SGD, since Adam is more commonly used for non-private language model fine-tuning. While the two algorithms differ in their parameter update rule, the basic gradient privatization procedure is the same. We performed additional experiments fine-tuning GPT-2 on the E2E dataset with DP-SGD (with freshly tuned learning rate and clipping norm values) and observed that its performance (test set BLEU 63.175 at $(\epsilon, \delta) = (8, 10^{-5})$) is on par with DP-Adam (test set BLEU 63.189 at $(\epsilon, \delta) = (8, 10^{-5})$).

## T    SUBTLETIES OF IMPLEMENTING DP MIXED PRECISION TRAINING

Mixed precision training (Micikevicius et al., 2017) accelerates updates by storing certain tensors in half-precision. To mitigate negative effects caused by potential arithmetic underflow, usual implementations upscale the loss with an adaptive factor pre-backpropagation and downscale the gradients with the same factor post-backpropagation. The scaling factor is adapted based on whether underflow is observed during training.

Special care needs to be taken when combining gradient privatization with mixed precision training. One implementation that ensures similar results across full and mixed precision training (1) upscales the loss with the adaptive factor $K$, (2) clips per-example gradients by $CK$, (3) adds to the sum of clipped gradients the usual Gaussian noise multiplied by $K$, and (4) downscales the noisy gradient by $K$. This is the implementation that we adopt, and we were able to obtain similar results on the E2E dataset with and without mixed precision.

One alternative implementation (a) upscales the loss with the adaptive factor $K$, (b) clips per-example gradients by $C$, (c) adds the usual Gaussian noise to the sum of clipped gradients, and (d) downscales noisy gradients by $K$. The main difference between this procedure and the prior is whether the factor

$K$ is considered during clipping and noising. This procedure, while having the same DP guarantee, typically does not result in similar results as full precision when the same hyperparameters are used across the two settings (even with an optimizer like Adam which self-adjusts the magnitude of updates with accumulated empirical second moments). We also identified that this implementation is also the primary reason that a prior work's code does not reproduce good results when run in full precision.[19]

## U  FINE-TUNING WITH RGP AND THE TEXT-INFILLING OBJECTIVE

Recall Section 3.2 and Table 1 showed that private full fine-tuning with a text-infilling objective leads to improved classification results. Here, we show that the infilling objective is also helpful when one privately fine-tunes with the RGP method for a classification task. To study this, we reimplemented the RGP method and tuned the hyperparameters in full precision. Fixing all hyperparameters, we compared models trained with and without infilling. Table 14 confirms that fine-tuning with RGP based on infilling is generally helpful across the considered tasks. Note the aim of this experiment is not obtain state-of-the-art performance, but rather to study the effect of using the text-infilling objective. Thus, we expect these results could generally be further improved with more extensive hyperparameter search.

Table 14: Text-infilling objective improves the performance of RGP for classification. Numbers are averaged over three independent runs.

| Method | $\epsilon = 8$ | | | |
| --- | --- | --- | --- | --- |
| | MNLI-(m/mm) | QQP | QNLI | SST-2 |
| RGP (RoBERTa-base) | 79.79/80.40 | 83.58 | 84.14 | 89.60 |
| RGP + infilling (RoBERTa-base) | 81.97/82.28 | 84.02 | 87.20 | 92.85 |

## V  WHICH METHOD SHALL I USE FOR PRIVATE FINE-TUNING?

While we were able to obtain similar results on the E2E and DART datasets with full fine-tuning and LoRA (Tables 1 and 8), we encountered significant difficulties in fine-tuning for dialog generation (even non-privately) when not updating the embedding layer and language modeling head – perplexity was much worse, and generations frequently contained seemingly arbitrary characters. This result suggests that while full fine-tuning may be more computationally intensive than lightweight approaches at times, its simplicity (involving few design decisions) makes it an attractive first option when compute resource is sufficient.

---

[19] https://github.com/dayu11/Differentially-Private-Deep-Learning

