# OpenReview forum: "Large Language Models Can Be Strong Differentially Private Learners"
_ICLR.cc/2022/Conference — ICLR 2022 Oral_

### Official Review · Reviewer_uqBn · 2021-11-02

**Correctness:** 3
**Technical Novelty And Significance:** 3
**Empirical Novelty And Significance:** 3
**Recommendation:** 8
**Confidence:** 4

**Main Review:**

Strengths:
1. As a DP algorithm on a deep learning model, this algorithm achieves good performance on fine-tuning the language models and several real-world applications.
2. The experimental results are sufficient and solid.
3. The performance gap between the proposed method and the non-private method is quite small, which shows applying the proposed DP algorithm would not decrease the utility so much.
4. Some of the conclusions from the experiments seem to be adaptive to other applications. For example, the relations among parameter size (for fine-tuning), privacy leakage, and performance.

Weaknesses:
1. The \delta in (\epsilon, \detla)-DP is too large for training a DP model with a meaningful level of privacy protection. Normally, \detla should be much smaller than the inversed dataset size (cannot be in the same scale of the inversed dataset size), some researchers choose \detla = 1/(|D| * log(|D|)), where |D| stands for the dataset size.

2. It would be better to conduct an ablation study to verify the promotion of the proposed methods (ghost clipping technique, fine-tuning on only a part of the parameters, and DP-adam). It's better to make it clear how much does each strategy contribute.

3. In this paper, the authors introduce a lot about the selection of hyper-parameters, thus hyper-parameters play an important role in model training. However, tuning hyper-parameters requires additional information about the private information (accessing the validation set or testing set), which leads to private leakage. So, how to count the information leakages of users? How to avoid those private leakages.

4. Is the DP-Adam proposed by yourself? If not, the corresponding reference and explanation are needed. Does it come from [Wang 2019]? If it is proposed by yourself, I suggest the author make it clear as it's one of the contributions.

[Wang 2019] DP-LSSGD: An Optimization Method to Lift the Utility in Privacy-Preserving ERM.

**Summary Of The Paper:**

In this paper, the authors propose a method that applies DP-SGD to NLP tasks. DP-SGD protects the privacy of the model training against that the individual information about the training samples is detected or inferred. The method is applied to the fine-tuning phase of the pre-trained language models (e.g. bert, gpt), thus it achieves good performances for many applications. To adapt DP-SGD to NLP models, this paper proposes ghost clipping that allows clipping in DP-SGD to run without instantiating per-example gradients for any layer in the model.

**Summary Of The Review:**

The paper is quite good on experiments and their possible applications but faces some questions.

---

> ### Author Response · Authors · 2021-11-14
> **Authors response (part 1)**
>
> We thank the reviewer for thoughtful comments. Below we address each concern separately.
>
> > 1) The \delta in (\epsilon, \detla)-DP is too large for training a DP model with a meaningful level of privacy protection.
>
> We agree with the reviewer that setting $\delta$ requires care. We’d like to emphasize that we use a $\delta$ value of $1 / (2 * |D_\text{train}|)$ across all experiments (see Section 2 of our initial draft). We recap several points we made while addressing reviewer E6VG’s concerns.
>
> First, a value as such is quite common in the existing literature. For instance, the seminal work by Abadi et al. [1] sets $\delta=1e-5$ for MNIST, where $|D_\text{train}|=60000$. Here, $\delta \approx 1 / (1.67 * |D_\text{train}|)$. Thus, compared to Abadi et al. [1], our values of $\delta$ are already much more conservative. Similar values of $\delta$ can also be found in more recent works [2, 3]. Note that the reference provided by the reviewer [4] also sets $\delta=1e-5$ for MNIST, which is much less conservative than our values.
>
> Second, we stress that the privacy guarantee for DP optimizers vary smoothly, i.e., approximate-DP holds for DP-SGD (or any DP optimizer) for a collection of $(\epsilon(\delta), \delta)$ pairs, where $\epsilon(\delta)$ is always finite and approaches infinity when $\delta\to 0$. This fact follows from the Renyi-DP to approximate-DP conversion [5]. The catastrophic failure mode with the name-and-shame mechanism -- an instance where the privacy loss random variable is consistently unbounded on the failure set of non-zero measure -- does not occur with DP-SGD (or any DP optimizer).
>
> Finally, we stress that the value of $\delta$ affects the noise multiplier in DP optimizers in only minor ways. We ran additional experiments to test this. For instance, on E2E, with epsilon=3, the noise multiplier goes from 1.225 with $\delta=1e-7$ to 1.156 with $\delta=1e-5$. The performance change is also minor. For instance, on E2E, test set BLEU is $58.01\pm1.88$ for $(\epsilon, \delta)=(3, 1e-7)$, and $58.95\pm1.76$ for $(\epsilon, \delta)=(3, 1e-5)$ over 5 independent runs.
>
> We thank the reviewer for raising this point, and will provide more clarification in the main text as well as include the new results in the Appendix.
>
> > 2) This paper did not compare to other related papers in the experiments. Comparing with the vanilla DP-SGD is a necessary baseline.
>
> The careful reader should see that for sentence classification tasks, we compared to the only previous paper [6] that attempted this task (results in Table 1). For language generation tasks E2E and DART, since we’re the first to perform experiments on these tasks under differential privacy, we compared to possible ways one could combine existing fine-tuning approaches with DP optimization (results in Table 2). The setup of these experiments are detailed carefully in Section 5 and Appendix K.
>
> During the author response period, we ran experiments with DP-SGD. We tuned the learning rate hyperparameter afresh for DP-SGD. On E2E, the best performing setup for $\epsilon=8$ (learning rate = 3, clipping norm = 0.1) attains a BLEU score of 63.17 (averaged over 3 seeds). **This is on par with the performance that DP-Adam obtains on this task.**
>
> We will include these results in the Appendix in the next revision.

---

> > ### Author Response · Authors · 2021-11-14
> > **Author response (part 2)**
> >
> > > 3) It would be better to conduct an ablation study to verify the promotion of the proposed methods (ghost clipping technique, fine-tuning on only a part of the parameters, and DP-adam). It's better to make it clear how much does each strategy contribute.
> >
> > Note that ghost clipping is a computational technique that enables clipping per-example gradients with low (GPU) memory cost -- this trick does not alter the privacy-utility tradeoff of the underlying DP optimization algorithm; one gets no more and no less privacy leakage with this technique compared to without it. We believe we made this very clear in our initial draft, but would revise to make it even clearer in the next draft.
> >
> > As for different ways of fine-tuning, we note that Table 2 already includes many of the established fine-tuning approaches, some of which optimizes parts of the model. For instance, `top2` optimizes the last 2 Transformer blocks of a large Transformer model, and `LoRA` only optimizes low-rank matrices which are added to the dense matrices of linear layers in a Transformer model. While we could include comparisons against more bizarre fine-tuning approaches (e.g., optimize only the first weight of each weight matrix), we have opted to only test against the most prominent examples in the literature (prefix-tuning, top2, LoRA, and RGP).
> >
> > Note that DP-Adam is quite well-known in the literature, and implementations/usages of it appear in almost all the popular differentially private machine learning libraries (e.g., DP-Adam in [tf-privacy](https://github.com/tensorflow/privacy/blob/7c4f5bab0964bd32b7ceafa009d9488920856440/tensorflow_privacy/privacy/optimizers/dp_optimizer.py#L385), [CIFAR-10 private training example in Opacus](https://github.com/pytorch/opacus/blob/9cda8072e52049a06afba7ab524276bb6613a727/examples/cifar10.py#L358), [DCGAN private training example in Opacus](https://github.com/pytorch/opacus/blob/9cda8072e52049a06afba7ab524276bb6613a727/examples/dcgan.py#L295)). We don’t claim this to be part of our contribution. Our contribution rather is applying this algorithm to a new setting (language model *full* fine-tuning), and showing that it works well when applied appropriately (our memory trick is one addition that makes the overall procedure computationally efficient). Note, even though we’re applying a known algorithm, we believe that applying it appropriately is highly non-trivial and that our empirical results are significant.
> >
> > Note that DP-Adam has the same gradient privatization procedure as DP-SGD (subsampled Gaussian mechanism). Any manipulation of the privatized gradient has the same privacy guarantee due to post-processing. Similarly, one may construct privatized versions of any known non-private optimizer, by first performing gradient privatization and then performing the usual update rule.
> >
> > Originally, the reason that we experimented with DP-Adam as opposed to DP-SGD was due to the popularity of Adam for non-private language model fine-tuning (one would agree to this claim if one has fine-tuned any language model with an established codebase, e.g., [HuggingFace’s tutorials](https://github.com/pytorch/opacus/blob/9cda8072e52049a06afba7ab524276bb6613a727/examples/dcgan.py#L295), [fairseq’s tutorial](https://github.com/pytorch/fairseq/blob/main/examples/rxf/README.md)). As a side note, we’d also want to emphasize that a past work that fine-tunes language models with DP-Adam falsely claims they fine-tune with DP-SGD [6] ([their code](https://github.com/dayu11/Differentially-Private-Deep-Learning/blob/d52db8acb1c9fa83cd23457bdc7240a3ee5310a0/language/bert/bert_code/run_exp.py#L104) uses Adam as the optimizer).
> >
> > As mentioned in the previous comment, we have performed experiments with DP-SGD during the response period and will include these results in the next revision.

---

> > > ### Author Response · Authors · 2021-11-14
> > > **Author response (part 3)**
> > >
> > > > 4) However, tuning hyper-parameters requires additional information about the private information (accessing the validation set or testing set), which leads to private leakage.
> > >
> > > We thank the reviewer for pointing this out. We are fully aware of the fact that tuning hyperparameter with a private training set (and possibly private validation set) could lead to additional privacy leakage. That’s why we only tuned hyperparameters on the E2E task, and reused these hyperparameters for all other tasks. While there’s extra privacy leakage on the E2E dataset, there isn’t any for the other tasks (which is the majority of tasks considered in the paper). In general, we find that good hyperparameters tend to transfer well.
> > >
> > > In our initial draft, this was carefully documented in Appendix B (“In addition, model selection from hyperparameter tuning on private training data could incur extra privacy leakage. We skip the step of private hyperparameter tuning (Liu & Talwar, 2019; Chaudhuri & Vinterbo, 2013; Papernot & Steinke, 2021) and instead perform tuning only on the E2E task and reuse almost the exact hyperparameters for all remaining tasks.”) and Appendix L.
> > >
> > > We will put this discussion in the main text for the next revision.
> > >
> > > > 5) Is the DP-Adam proposed by yourself? If not, the corresponding reference and explanation are needed. Does it come from [Wang 2019]? If it is proposed by yourself, I suggest the author make it clear as it's one of the contributions.
> > >
> > > As mentioned in our response to 3), DP-Adam is well-known in the literature, and implementations/usages of it occur in most of the popular private machine learning libraries. More clarification is given in our response to 3). We will make these points clearer in the next revision.
> > >
> > > [1] Abadi, M., Chu, A., Goodfellow, I., McMahan, H. B., Mironov, I., Talwar, K., & Zhang, L. (2016, October). Deep learning with differential privacy. In Proceedings of the 2016 ACM SIGSAC conference on computer and communications security (pp. 308-318).
> > >
> > > [2] Tramèr, F., & Boneh, D. (2020). Differentially private learning needs better features (or much more data). arXiv preprint arXiv:2011.11660.
> > >
> > > [3] Yu, D., Zhang, H., Chen, W., & Liu, T. Y. (2021). Do not let privacy overbill utility: Gradient embedding perturbation for private learning. arXiv preprint arXiv:2102.12677.
> > >
> > > [4] Wang, B., Gu, Q., Boedihardjo, M., Barekat, F., & Osher, S. J. (2019). DP-LSSGD: An Optimization Method to Lift the Utility in Privacy-Preserving ERM.
> > >
> > > [5] Mironov, I. (2017, August). Rényi differential privacy. In 2017 IEEE 30th Computer Security Foundations Symposium (CSF) (pp. 263-275). IEEE.
> > >
> > > [6] Yu, D., Zhang, H., Chen, W., Yin, J., & Liu, T. Y. (2021). Large Scale Private Learning via Low-rank Reparametrization. arXiv preprint arXiv:2106.09352.

---

> > > > ### Comment · Reviewer_uqBn · 2021-11-15
> > > > **Thanks for authors' response**
> > > >
> > > > The response from the authors solved most of my concerns. I hope the authors make those points more clear in the next version. I've also read the comments from other reviewers. Thanks for the authors' rebuttal and other reviewers' comments. I prefer to increase my score.
> > > >
> > > > BTW, when will the source code be available for the public?

---

> > > > > ### Author Response · Authors · 2021-11-15
> > > > > **Thanks**
> > > > >
> > > > > We thank the reviewer for spending their time reading through our comments.
> > > > >
> > > > > Link to our code will be posted after the double-blind review period.

---

### Official Review · Reviewer_E6VG · 2021-11-02

**Correctness:** 3
**Technical Novelty And Significance:** 3
**Empirical Novelty And Significance:** 4
**Recommendation:** 6
**Confidence:** 4

**Main Review:**

Pros,
1) It achieves a remarkable performance of DP algorithms on NLP tasks with a satisfactory level of privacy protection.
2) There will be a number of potential applications since this paper verify the model on multiple NLU and NLG tasks.
3) It works on several pre-trained language models (e.g. GPT, BERT, Roberta).
4) The experiments part is quite solid and covers many necessary details.

Cons,
1) The experimental results in table 1~3 compared to the non-private seems amazing, however, the comparison with the baseline methods is also required, which would show the contribution of your proposed methods. There're several possible baselines: directly apply DP-SGD and an application of DP-SGD on language models (GPT-2) [1].

2) The choice of \epsilon makes sense because the \epsilon in a range of 0.1~5 provides meaningful protection. However, the choice of \delta in experiments may cause some problems in privacy leakage. The value of \delta on the order of 1/|D_train| is very dangerous according to [2]. In that level of \delta, they permit “preserving privacy” by publishing the complete records of a small number of database participants. It's better to choose the \delta as 1/( |D_train| *100), which may reduce the utility but ensure the privacy is well protected.

3) Will the ghost clipping increases the utility or privacy protection? Could you please provide some insights (or even quantitative analysis) about it?

4) It would be better to conduct some key experiments with a range of \espilon from 0.1 to 10 (e.g. 0.1, 0.5, 1, 2, 3, 5, 8). Many DP learning algorithms use a curve to show the variation tendency of utilities for the different \epsilon.

5) This paper did a good job on the empirical study, but the technical novelty is limited.

[1] Differentially Private Language Models Benefit from Public Pre-training. 2020

[2] The Algorithmic Foundations of Differential Privacy, 2014


After rebuttal:
Thanks for your explanation! I believe it's a good paper and I'm happy to see its acceptance.


**Summary Of The Paper:**

This paper adapts the widely used DP learning algorithm, DP-SGD, to language models. It achieves to fine-tune the dataset while protecting the private information in the dataset. In this paper, the authors conduct some empirical studies on language models and find some useful conclusions (e.g. fine-tuning on a part of parameters with DP is enough). The authors verify the model on sentence classification, table-to-text generation, and dialog generation tasks, using various pre-trained language models (e.g. GPT, Bert).

**Summary Of The Review:**

The paper has a clear motivation and the main idea is also interesting, but still suffers from some issues.

---

> ### Author Response · Authors · 2021-11-14
> **Author response (part 1)**
>
> We thank the reviewer for their thoughtful comments. Below we address each concern separately.
>
> > The experimental results in table 1~3 compared to the non-private seems amazing, however, the comparison with the baseline methods is also required, which would show the contribution of your proposed methods. There're several possible baselines: directly apply DP-SGD and an application of DP-SGD on language models (GPT-2) [1].
>
> Recall that we perform full fine-tuning with DP-Adam. Note that DP-Adam is quite well-known in the literature, and implementations/usages of it appear in almost all the popular differentially private machine learning libraries (e.g., DP-Adam in [tf-privacy]( https://github.com/tensorflow/privacy/blob/7c4f5bab0964bd32b7ceafa009d9488920856440/tensorflow_privacy/privacy/optimizers/dp_optimizer.py#L385), [CIFAR-10 private training example in Opacus](https://github.com/pytorch/opacus/blob/9cda8072e52049a06afba7ab524276bb6613a727/examples/cifar10.py#L358), [DCGAN private training example in Opacus](https://github.com/pytorch/opacus/blob/9cda8072e52049a06afba7ab524276bb6613a727/examples/dcgan.py#L295)). We don’t claim this to be part of our contribution. Our contribution rather is applying this algorithm to a new setting (language model *full* fine-tuning), and showing that it works well when applied appropriately (our memory trick is one addition that makes the overall procedure computationally efficient). Note, even though we’re applying a known algorithm, we believe that applying it appropriately is highly non-trivial and that our empirical results are significant.
>
> Note that DP-Adam has the same gradient privatization procedure as DP-SGD (subsampled Gaussian mechanism). Any manipulation of the privatized gradient has the same privacy guarantee due to post-processing. Similarly, one may construct privatized versions of any known non-private optimizer, by first performing gradient privatization and then performing the usual update rule.
>
> Originally, the reason that we experimented with DP-Adam as opposed to DP-SGD was due to the popularity of Adam for non-private language model fine-tuning (one would agree to this claim if one has fine-tuned any language model with an established codebase, e.g., [HuggingFace’s tutorials](https://huggingface.co/transformers/training.html), [fairseq’s tutorial](https://github.com/pytorch/fairseq/blob/main/examples/rxf/README.md)). As a side note, we’d also want to emphasize that a past work that fine-tunes language models with DP-Adam falsely claims they fine-tune with DP-SGD [1] ([their code](https://github.com/dayu11/Differentially-Private-Deep-Learning/blob/d52db8acb1c9fa83cd23457bdc7240a3ee5310a0/language/bert/bert_code/run_exp.py#L104) uses Adam as the optimizer).
>
> To directly address the reviewer’s concern, we ran additional experiments with DP-SGD. We tuned the learning rate hyperparameter afresh for DP-SGD. On E2E, the best performing setup for $\epsilon=8$ (learning rate = 3, clipping norm = 0.1) attains a BLEU score of 63.17 (averaged over 3 seeds). **This is on par with the performance that DP-Adam obtains on this task.**
>
> We stress that the reference which the reviewer cited [2] does not at all fine-tune GPT-2 with DP-SGD. The careful reader should notice that they only fine-tune GPT-2 **non-privately**. For their private experiments, they fine-tuned a feedforward network that **they pretrained themselves** (see their Section 4.3 for the details). Since they didn’t privately fine-tune a high quality pretrained model like GPT-2, their empirical finding was that even with fine-tuning, DP models are orders of magnitude worse than state-of-the-art non-private models (see Section 5 -- “The perplexity scores for both the small and large feedforward language models are orders of magnitude worse than the GPT-2 models indicating that they are not competitive with state of the art language models.”). This empirical finding is much the opposite of our conclusions.
>
> We thank the reviewer again for raising these subtle points, and we will make all of these points clear in the next revision.

---

> > ### Author Response · Authors · 2021-11-14
> > **Author response (part 2)**
> >
> > > ​​However, the choice of \delta in experiments may cause some problems in privacy leakage. The value of \delta on the order of 1/|D_train| is very dangerous according to [2]. In that level of \delta, they permit “preserving privacy” by publishing the complete records of a small number of database participants. It's better to choose the \delta as 1/( |D_train| *100), which may reduce the utility but ensure the privacy is well protected.
> >
> > While we agree that how $\delta$ should be set requires much thought, we’d like to emphasize that we use a $\delta$ value of $1 / (2 * |D_\text{train}|)$ across all experiments (see Section 2 of our draft).
> >
> > First, a value as such is quite common in the existing literature. For instance, the seminal work by Abadi et al. [3] sets $\delta=1e-5$ for MNIST, where $|D_\text{train}|=60000$. Here, $\delta \approx 1 / (1.67 * |D_\text{train}|)$. Thus, compared to Abadi et al. [3], our values of $\delta$ are already much more conservative. Similar values of $\delta$ can also be found in more recent works [4, 5].
> >
> > Second, we stress that the privacy guarantee for DP optimizers vary smoothly, i.e., approximate-DP holds for DP-SGD (or any DP optimizer) for a collection of $(\epsilon(\delta), \delta)$ pairs, where $\epsilon(\delta)$ is always finite and approaches infinity when $\delta\to 0$. This fact follows from the Renyi-DP to approximate-DP conversion [6]. The catastrophic failure mode with the name-and-shame mechanism -- an instance where the privacy loss random variable is unbounded on the failure set -- does not occur with DP-SGD (or any DP optimizer).
> >
> > Finally, we stress that the value of $\delta$ affects the noise multiplier in DP optimizers in only minor ways. We ran additional experiments to test this. For instance, on E2E, with epsilon=3, the noise multiplier goes from 1.225 with $\delta=1e-7$ to 1.156 with $\delta=1e-5$. The performance change is also minor. For instance, on E2E, test set BLEU is $58.01\pm1.88$ for $(\epsilon, \delta)=(3, 1e-7)$, and $58.95\pm1.76$ for $(\epsilon, \delta)=(3, 1e-5)$ over 5 independent runs.
> >
> > > The choice of \epsilon makes sense because the \epsilon in a range of 0.1~5 provides meaningful protection.
> >
> > We want to stress that there are many real-world deployments of DP with $\epsilon$ much larger than 5, with the US census being a prime example (see [these notes](https://www.census.gov/newsroom/press-releases/2021/2020-census-key-parameters.html)).
> >
> > > Will the ghost clipping increases the utility or privacy protection?
> >
> > As described in the paper, ghost clipping is a memory-saving trick for clipping gradients -- it does not alter the core algorithmic procedure of DP optimization. Assuming one has unlimited GPU memory, one should get similar results with and without ghost clipping. One should get the same privacy guarantee with and without ghost clipping, once all other hyperparameters are fixed (e.g., sampling rate, noise multiplier, number of updates).
> >
> > In practice, ghost clipping enables us to use larger (and higher quality) pretrained models that typically wouldn’t fit in memory otherwise. In this sense, ghost clipping helps us achieve better performance.
> >
> > > It would be better to conduct some key experiments with a range of \espilon from 0.1 to 10 (e.g. 0.1, 0.5, 1, 2, 3, 5, 8).
> >
> > We thank the reviewer for raising this point. We have run additional experiments on E2E. When $\epsilon < 2$, we obtain considerably worse results (BLEU score below 30). The test set performance change from $\epsilon=2$ to $\epsilon=8$ is more gradual. We will include these additional results in the Appendix.
> >
> > > This paper did a good job on the empirical study, but the technical novelty is limited.
> >
> > While papers that propose fancy techniques may seem interesting and eyecatching, we believe that the simplicity demonstrated in our work is a boon rather than a bane (not to mention it works well!). We believe that the simplicity of the approach is especially desirable in the realm of private learning since any additional design decision would imply more tuning -- hyperparameter tuning incurs privacy cost unless good hyperparameters can be transferred across tasks and domains.
> >
> > We believe we have adequately addressed the concerns raised by the reviewer, so we’d like to politely ask the reviewer to reconsider their assessment of our work.

---

> > > ### Author Response · Authors · 2021-11-14
> > > **Authors response (part 3)**
> > >
> > > [1] Yu, D., Zhang, H., Chen, W., Yin, J., & Liu, T. Y. (2021). Large Scale Private Learning via Low-rank Reparametrization. arXiv preprint arXiv:2106.09352.
> > >
> > > [2] Kerrigan, G., Slack, D., & Tuyls, J. (2020). Differentially private language models benefit from public pre-training. arXiv preprint arXiv:2009.05886.
> > >
> > > [3] Abadi, M., Chu, A., Goodfellow, I., McMahan, H. B., Mironov, I., Talwar, K., & Zhang, L. (2016, October). Deep learning with differential privacy. In Proceedings of the 2016 ACM SIGSAC conference on computer and communications security (pp. 308-318).
> > >
> > > [4] Tramèr, F., & Boneh, D. (2020). Differentially private learning needs better features (or much more data). arXiv preprint arXiv:2011.11660.\
> > >
> > > [5] Yu, D., Zhang, H., Chen, W., & Liu, T. Y. (2021). Do not let privacy overbill utility: Gradient embedding perturbation for private learning. arXiv preprint arXiv:2102.12677.
> > >
> > > [6] Mironov, I. (2017, August). Rényi differential privacy. In 2017 IEEE 30th Computer Security Foundations Symposium (CSF) (pp. 263-275). IEEE.

---

### Official Review · Reviewer_a9ro · 2021-11-03

**Correctness:** 3
**Technical Novelty And Significance:** 2
**Empirical Novelty And Significance:** 3
**Recommendation:** 8
**Confidence:** 3

**Main Review:**

Overall, this is a fairly empirical paper on an important problem and shows good performance. The authors presented a set of thorough experiments investigating the impact of various hyper-parameters, which are widely known as sensitive and difficult to tune, providing a nice and informative guidelines for other researchers and practitioners who work in this area. The ghost clipping trick proposed in this paper for memory reducing in DP-SGD is simple yet effective, greatly reducing the memory cost when applying DP-SGD to large models, especially the popular large-scale pre-trained language models, potentially encouraging more research effort in the DP learning area.

The experiments presented in this paper are quite solid and clear, both well-designed and documented, and most of the claims made in the paper are reasonable and well-supported. My only complaint is that I wish to see more explanations and more principled approach to selecting the hyper-parameters in fine-tuning with DP-SGD, but I guess it is out of this paper's scope (the authors did provide some empirical discussions in the paper and the appendix, which I appreciate).

**Summary Of The Paper:**

This paper investigated the problem of privately fine-tuning large language models for downstream NLP tasks, including sentence classification and language generation. The authors showed that by appropriately selecting hyper-parameters (including batch size, learning rate, training epochs, and clipping norm) and making the fine-tuning task aligned with pretraining tasks, directly fine-tuning large language models with DP-SGD yields strong performance, and provided an empirical guideline for setting a good training configuration. The authors also proposed ghost clipping trick for further memory saving when fine-tuning large language models. Finally the authors showed through experiments that low dimensional updates do not necessarily lead to better performance.

**Summary Of The Review:**

This paper presented a detailed investigation on the training configurations of fine-tuning large language models with DP-SGD through a set of extensive experiments, which are beneficial to researchers and practitioners working on private NLP, and the proposed ghost clipping trick for DP-SGD greatly reduces memory when applied to large language models, making private NLP research more feasible. I believe both the insights and the proposed method in this paper will bring value to the private NLP community.

---

> ### Author Response · Authors · 2021-11-10
> **Response**
>
> We thank the reviewer for their thoughtful comments and their positive sentiment of our work. Below, we address the concern of the reviewer.
>
> > My only complaint is that I wish to see more explanations and more principled approach to selecting the hyper-parameters in fine-tuning with DP-SGD, but I guess it is out of this paper's scope (the authors did provide some empirical discussions in the paper and the appendix, which I appreciate).
>
> We share the view with the reviewer that perhaps more can be done with regards to hyperparameter selection. While we believe that hyperparameter tuning and good hyperparameters are ultimately task-dependent, experiments in our paper have demonstrated very encouraging results with hyperparameter transfer.
>
> Notably, we only tuned hyperparameters on the E2E task in the paper and reused almost the exact hyperparameter set for other tasks (e.g., sentence classification, dialog generation, and DART).  The details of how we transferred the hyperparameters are documented in Appendix B and Appendix L.
>
> We believe that hyperparameter transfer (with tuning performed on public data) is a practical approach to bypassing the compute cost associated with per-task tuning and the privacy leakage caused by tuning on private training data.
>
> We will include the discussion on hyperparameter transfer in the main text in the next revision.

---

### Official Review · Reviewer_XPKW · 2021-11-05

**Correctness:** 3
**Technical Novelty And Significance:** 4
**Empirical Novelty And Significance:** 3
**Recommendation:** 8
**Confidence:** 5

**Main Review:**

Strength of the paper:
1. The paper is clearly written and easy to understand. It presents the DP training problem very well and demonstrates the challenge for large pretrained NLP models.
2. The proposed GhostClipping is effective in reduce memory consumption, at (not too big) cost of an extra backpropagation. The method is very clear and also simple to implement.
3. The experiments show strong results on both classification and text generation tasks, better than the prior DP methods in terms of memory consumption and accuracy/generation quality.

Weakness of the paper:
1. There are three aspects involved. GhostClipping, large batch/proper learning rate, fine-tuning with masked prediction. It seems the accuracy on text classification comes from the later two. Table 1, comparing full(RoBERTa-large) with RGP(RoBERTa-large) does not show improvement (please list the average score here). Full is actually worse. If RGP is augmented with larger batch/proper learning rate, and masked prediction for fine-tuning, it may be the best in terms of accuracy.
2. Figure 1 is a bit mis-leading. Please show and compare with RoBERTa (non-private) on 1(a) and non-private GPT-2 on 1(b). Those are the baselines to compare. Otherwise you are comparing private RoBERTa with non-private BERT or non-private T-GEN(non Transformer). Statement in the abstract and introduction are over-stated.
3. The organization of the paper could be improved. Ghost-Cliping is your main method, which could be moved earlier.


**Summary Of The Paper:**

The paper propose a faster algorithm to learn approximate differentially private NLP models. Pretrained NLP models are often very large. Practical procedure involves fine-tuning NLP model on private data, which may leak private information. To avoid leakage, DP-SGD (and DP-AGAGRAD, DP-ADAM) uses norm clipping on each sample’s gradient, and then add isotropic noise to aggregated gradients for samples in a batch. Then the normal update steps of SGD, ADAGRAD, or ADAM are performed. This will ensure privacy under differential privacy definition. However, this procedure requires computing per-sample gradient and keep them in the memory so that the norm of the gradient can be calculated, and the per-sample gradient clipping can be performed. This will introduce memory overhead proportional to the batch_size$\times$#params, which is impossible for very large NLP models.

This paper propose GhostClipping method to save the memory, without the need of per-sample gradient instantiated. The idea is compute the partial sum of gradient element-wise square using two small matrices of the size of $T\times T$, where $T$ is sequence length (<=1024 in practice), and aggregate them to obtain the per-sample norm (just a scale for each). And then it performs a second back-propagation to compute aggregation on the clipped gradient. It will uses almost the same memory as standard SGD (or ADAM), with one forward pass and two backward passes.

This paper also introduce two additional techniques to improve, one is choose a larger batch size, and the other is to introduce multi-task finetuning (fine-tuning includes masked prediction task on the target dataset). These two are important in boost the performance of the final models.

The paper evaluates on text classification, data-to-text generation, and dialog generation tasks. The results shows it gains prior DP methods.





**Summary Of The Review:**

Reason to accept:
Simple DP algorithm with demonstrated reduction in memory consumption and effectiveness. It enables large pre-trained NLP models to be further fine-tuned on private data under approximate DP. Clear presentation.

Reason to reject:
Better to identify the cause of improvement and make a fair comparison to baseline models (in particular for text classification tasks). Over-statement and misleading in abstract and introduction about comparable to strong non-private baselines.

---

> ### Author Response · Authors · 2021-11-15
> **Author response (part 1)**
>
> We thank the reviewer for their thoughtful comments. Below we address each concern.
> > If RGP is augmented with larger batch/proper learning rate, and masked prediction for fine-tuning, it may be the best in terms of accuracy.
> The results we include for RGP are from [their released codebase](https://github.com/dayu11/Differentially-Private-Deep-Learning/tree/main/language), and using the tuned hyperparameters from their original work.
>
> We stress that we **did not** tune hyperparameters for any of the classification tasks, but rather used hyperparameters transferred from the E2E data-to-text generation task. This was to avoid incurring extra privacy leakage due to hyperparameter tuning. This was documented in Appendix B and Appendix L of our initial draft (“In addition, model selection from hyperparameter tuning on private training data could incur extra privacy leakage. We skip the step of private hyperparameter tuning (Liu & Talwar, 2019; Chaudhuri & Vinterbo, 2013) and instead perform tuning only on the E2E task and reuse almost the exact hyperparameters for all remaining tasks.”).
>
> While it may seem fair to compare well-tuned full vs well-tuned RGP, we generally prefer to avoid large-scale tuning due to the potential of increased privacy leakage. Note that this matter of privacy cost incurred by tuning with private training (and potentially private validation) data has also been raised by other reviewers (a9ro, uqBn).
>
> In addition, it’s important to note that the tuning strategies that we outlined in Section 3.1 do not necessarily transfer to the RGP setting. For instance, using small clipping norms (under mixed-precision) typically isn’t better for RGP from our experiments. This is likely an artifact of RGP not directly privatizing gradients -- rather, for a parameter weight matrix, the method runs power iteration to obtain rank-k tiles (k=1 in most cases), for which gradients are computed.
>
> To test if the mask-infilling objective improves performance for RGP, we ran additional ablation studies comparing RGP with and without the mask-infilling objective. On SST-2 for $\epsilon=8$ with RoBERTa base, validation accuracy improves from $90.64\pm 0.26$ to $92.91\pm 0.40$ over 5 independent runs.
>
> Since the released code for RGP fails when running in full precision, we re-tuned hyperparameters of our re-implementation in full precision with our mask-infilling objective (for training stability purposes). The above results we report were obtained with this tuned hyperparameter set (thus the absolute numbers aren’t strictly comparable to our non-tuned full fine-tuning numbers).
>
> We thank the reviewer again for raising these points. We will include additional results and discussions based on this thread in the next revision.

---

> > ### Author Response · Authors · 2021-11-15
> > **Author response (part 2)**
> >
> > > Figure 1 is a bit mis-leading. Please show and compare with RoBERTa (non-private) on 1(a) and non-private GPT-2 on 1(b). Those are the baselines to compare. Otherwise you are comparing private RoBERTa with non-private BERT or non-private T-GEN(non Transformer). Statement in the abstract and introduction are over-stated.
> >
> > We thank the reviewer for these suggestions. We’ll include the numbers for non-private RoBERTa and GPT-2 in the next revision.
> >
> > Measuring the performance drop of the method of full fine-tuning private vs non-private is important. However, this wasn’t meant to be the main message of Figure 1. Our original intent was not to suggest that private learning has no cost on utility. Rather, we intended to convey the idea that **direct DP full fine-tuning using increasingly better pretrained models eventually attains the level of non-private performance that was considered state-of-the-art a few years ago**. This is very encouraging from a practical standpoint, especially considering that enforcing DP in NLP problems was considered quite impractical by many in the field. The appeal also lies in the simplicity of our demonstrated approach -- the most basic method works well, and more progress can be expected in the future as pretraining consistently improves from existing developments.
> >
> > Perhaps another way to state the above is that the “introducing new methodology” aspect is only one part of our paper among many others. We believe that our paper is a combination of empirical analyses (e.g., Section 3.1), empirical evaluation/benchmarking (e.g., Section 5.2 & Table 2), and new methodology (e.g., Section 4). We also believe that all aspects are indispensable components of the paper, each of which either provides explanations of why things should work or provides guidelines on how to make things run efficiently in practice.
> >
> > In general, we also don’t believe that every paper in the machine learning community has been or should be an “introduce-a-new-methodology-type” paper. Recent works on empirical evaluation and analysis, and benchmarking have revealed great insights that pushed the field forward and led to tangible progress [1, 2, 3, 4, 5, 6].
> >
> > > The organization of the paper could be improved. Ghost-Cliping is your main method, which could be moved earlier.
> >
> > We share the sentiment with the reviewer that there’s room for improvement in terms of organization. Overall, we think that ghost clipping is only one part of our paper. How to get full fine-tuning to work well (Section 3), and does dimensionality degrade performance (Section 5) are important aspects that we deem necessary to gain a better understanding of DP language model fine-tuning.
> >
> > We will improve the organization by updating parts of the introduction.
> >
> > [1] Schmidt, Robin M., Frank Schneider, and Philipp Hennig. "Descending through a crowded valley-benchmarking deep learning optimizers." International Conference on Machine Learning. PMLR, 2021.
> >
> > [2] Choi, D., Shallue, C. J., Nado, Z., Lee, J., Maddison, C. J., & Dahl, G. E. (2019). On empirical comparisons of optimizers for deep learning. arXiv preprint arXiv:1910.05446.
> >
> > [3] Zhang, G., Li, L., Nado, Z., Martens, J., Sachdeva, S., Dahl, G., ... & Grosse, R. B. (2019). Which algorithmic choices matter at which batch sizes? insights from a noisy quadratic model. Advances in neural information processing systems, 32, 8196-8207.
> >
> > [4] Shallue, C. J., Lee, J., Antognini, J., Sohl-Dickstein, J., Frostig, R., & Dahl, G. E. (2018). Measuring the effects of data parallelism on neural network training. Journal of Machine Learning Research 20 (2019) 1-49.
> >
> > [5] Kaplan, J., McCandlish, S., Henighan, T., Brown, T. B., Chess, B., Child, R., ... & Amodei, D. (2020). Scaling laws for neural language models. arXiv preprint arXiv:2001.08361.
> >
> > [6] ​​Brown, T. B., Mann, B., Ryder, N., Subbiah, M., Kaplan, J., Dhariwal, P., ... & Amodei, D. (2020). Language models are few-shot learners. arXiv preprint arXiv:2005.14165.

---

### Author Response · Authors · 2021-11-21
**Rebuttal revision**

We have updated our draft based on suggestions from reviewers.

To summarize the most important aspects:
- We provided more background about DP-Adam in Appendix A and included results comparing DP-SGD against DP-Adam in the new section Appendix S.
- We included additional results showing that varying $\delta$ only affects performance marginally in the new section Appendix R. This section also has results of the effect on performance when varying $\epsilon$.
- We included non-private numbers of GPT-2 and RoBERTa in Figure 1.
- We outlined subtleties in DP mixed-precision training in the new section Appendix T.
- We included experimental results comparing RGP with and without using the text-infilling objective in the new section Appendix U.
- We included additional discussion on difficulties in fine-tuning when the embedding and language modeling heads aren't updated for dialog generation in the new section Appendix V.
- We modified the writing of the abstract and introduction to be more precise. We provided more details about the implementation of the memory-saving technique in Section 4.

We thank the reviewers again for their time and valuable suggestions!

---

### Decision · Program_Chairs · 2022-01-20

**Decision:**

Accept (Oral)

**Comment:**

This work adapts the widely used DP learning algorithm to language models. Reviewers all agreed that this work tackles an important problem with clear motivation and thorough experiments, and achieved strong performance (memory reduction and effectiveness) on NLP tasks.  Thus, we recommend an acceptance.